# Benchmark of Benchmarks: Unpacking Influence and Code Repository Quality in LLM Safety Benchmarks

## Abstract

The rapid growth of research in LLM safety makes it hard to track all advances. Benchmarks are therefore crucial for capturing key trends and enabling systematic comparisons. Yet, it remains unclear why certain benchmarks gain prominence, and no systematic assessment has been conducted on their academic influence or code quality. This paper fills this gap by presenting the first multi-dimensional evaluation of the influence (based on five metrics) and code quality (based on both automated and human assessment) on LLM safety benchmarks, analyzing 31 benchmarks and 382 non-benchmarks across prompt injection, jailbreak, and hallucination. We find that benchmark papers show no significant advantage in academic influence (e.g., citation count and density) over non-benchmark papers. We uncover a key misalignment: while author prominence correlates with paper influence, neither author prominence nor paper influence shows a significant correlation with code quality. Our results also indicate substantial room for improvement in code and supplementary materials: only 39% of repositories are ready-to-use, 16% include flawless installation guides, and a mere 6% address ethical considerations. Given that the work of prominent researchers tends to attract greater attention, they need to lead the effort in setting higher standards.

## 1 Introduction

The field of LLM safety is experiencing an arms race, with new attack techniques and defensive measures (Rawte et al., 2023; Ji et al., 2023; Li et al., 2023b; Chu et al., 2025b; Perez & Ribeiro, 2022; Greshake et al., 2023; Touvron et al., 2023; Ouyang et al., 2022; Chu et al., 2025a) being proposed constantly. Within two years of ChatGPT's launch (OpenAI, 2022), this dynamic has spurred the publication of over 250,000 new papers about LLMs, with nearly 50,000 on safety.[1]

The rapid growth of research in this area brings challenges in tracking the latest advancements. Benchmark studies play a pivotal role in helping catch the advancements by comparing research findings and providing valuable insights. Many benchmarks cover similar topics (e.g., jailbreak) and test overlapping models and datasets, yet some gain more traction than others for unclear reasons. Additionally, code repository quality and usability in benchmarks remain underexplored. Addressing these gaps is crucial for improving LLM safety research and ensuring effective evaluations.

***Research Questions.*** Our study centers on the three research questions (RQs) below to address the limitations: **[RQ1]** How significant is the influence of the current benchmark papers, and what factors relate to their influence? **[RQ2]** What is the quality of the code repositories associated with benchmark papers, and which factors relate to their quality? **[RQ3]** What is the relationship between the influence of benchmark papers and the quality of their associated code repositories?

We focus on three newly emerging key LLM safety topics: prompt injection, jailbreak, and hallucination. The overview of our data collection pipeline is shown in Figure 1. Specifically, we collect benchmark papers on the three topics through keyword searches in Semantic Scholar (Semantic Scholar, 2024) and Google Scholar (Google, 2024), with non-benchmark papers as a control group from four curated LLM safety paper repositories (ThuCCSLab, 2024; Zhou, 2024; Sitawarin, 2024;

---

[1]Statistics from Semantic Scholar (Figure 9 of Appendix P).

Figure 1: Data collection pipeline.

Table 1: Summary of the Data Under Study

| | Benchmark | Non-Benchmark |
|---|---|---|
| Collection Period | [November 30, 2022 – November 1, 2024] | |
| # Papers | 31 | 382 |
| # Repositories | 27 | 168 |
| # Jailbreak | 12 | 246 |
| # Hallucination | 13 | 103 |
| # Prompt Injection | 6 | 33 |

Corca AI, 2024) (Section 2.2). We further rely on the Semantic Scholar API to obtain the metadata (e.g., citation counts and author metrics) of these papers, the Papers with Code API (Meta AI, 2024) to link papers to their code repositories, and GitHub API (GitHub, 2024) to gather additional repository metadata (e.g., commit history and availability duration).

Covering research from November 30, 2022 (ChatGPT's launch) to November 1, 2024, we have built a dataset that includes 31 benchmark papers (with 27 public repositories) and 382 non-benchmark papers (the control group, with 168 public repositories). The data summary is in Table 1.

***Our Approach.*** Our analysis integrates descriptive methods, which provide an overview of paper influence and code quality, with inferential methods, which rigorously assess significance across comparisons and associations (Ostle, 1963; Martin & Altman, 2003). First, we assess the influence of benchmark papers using multiple metrics, including citation density and GitHub star density.[2] We also explore potential factors related to influence, such as the paper's affiliated area and the lead author's h-index. Then, we use both tool-based (based on Pylint and Radon) and human-based methods to assess code quality and explore factors that may be associated with it. Finally, we investigate the correlation between benchmark influence and code repository quality.

***Main Findings.*** We outline the following findings about LLM safety benchmark papers:

- (Section 4) Benchmark papers do not show a statistically significant difference in citation metrics compared to non-benchmark papers, but their code repository quality tends to be higher. Statistical associations are observed between several author-prominence metrics and influence metrics, for example, between Author H-Index (Top-1) and Citation Count.

- (Section 5) Both the code quality and the supplementary material quality have considerable room for improvement: only 39% of code repositories can run smoothly without modifications, only 16% provide flawless install guides, and only 6% include ethical considerations. Author prominence has no statistically significant correlation to code quality.[3]

- (Section 6) Simply providing code is not enough: we find no statistically significant difference in citation density between papers with code requiring modifications and those without any code. We also fail to find a significant correlation between a paper's citation density and the intrinsic code characteristics, such as static metrics or maintenance frequency.

## 2 DATA UNDER STUDY

### 2.1 SCOPE OF THIS STUDY

In this paper, we focus on three novel safety risks specific to LLMs, including **prompt injection** (Perez & Ribeiro, 2022; Greshake et al., 2023), **jailbreak** (Zou et al., 2023; Shen et al., 2023; Chu et al., 2025b), and **hallucination** (Rawte et al., 2023; Ji et al., 2023; Li et al., 2023b). These newly emerging safety topics in the LLM era pose common and serious real-world risks, offering us unique research opportunities. More details about why these topics are chosen are in Appendix B.

---

[2] Average citations or stars per day since public release.

[3] All related adjusted p-values ($p$) exceed threshold 0.05 (Fisher, 1970).

## 2.2 DATA COLLECTION

***Data Sources.*** The metadata of papers sources from Semantic Scholar (Semantic Scholar, 2024) and Google Scholar (Google, 2024). We retrieve the paper-repository pairs from Paper with Code (Meta AI, 2024). The metadata related to repositories, such as the GitHub star count, is collected from GitHub API (GitHub, 2024). Details of these data sources are introduced in Appendix E and the details of metadata are available in Appendix G.

***Benchmark Papers.*** In order to conduct a comprehensive compilation of benchmark papers across selected safety topics, we implement a systematic collection methodology utilizing three distinct keyword sets (detailed in Appendix C). Our data collection process leverages both the Semantic Scholar API and the Google Scholar platform to attempt comprehensive coverage of the literature on LLM safety. For each safety topic, we first query the Semantic Scholar API with the corresponding keyword set to search for related papers, targeting both titles and abstracts. The results are then complemented by direct searches using the same keyword sets on the Google Scholar website. Notably, our findings reveal a substantial overlap between results from both platforms (only two papers from each source that do not overlap), aligning with those from previous studies (Hannousse, 2021). We combine all overlapping and non-overlapping papers (in total 39 papers) for further inspection.

Subsequently, we conduct a rigorous manual inspection process to check if the collected benchmark papers meet both relevancy and quality criteria. Specifically, we filter papers according to the following three criteria: **[1]** The paper must be relevant to LLM safety and should focus on the selected safety topic, rather than matching keywords in unrelated fields (e.g., "iOS jailbreak"). **[2]** The paper should not be a SoK or survey paper; we retain only primary research papers rather than higher-level overview or position works. **[3]** The paper must be a genuine benchmark study, as evidenced by its research content, rather than merely referencing benchmark datasets in its methodology (e.g., "We evaluate our method on benchmark datasets."). The PRISMA diagram is shown in Figure 4 of Appendix F. *After the inspection, we keep 31 inspected benchmark papers.*

***Non-Benchmark Papers.*** To facilitate comparative analysis and assess the influence of benchmark papers, we additionally collect non-benchmark papers within the same safety topics to serve as control groups. We initially experimented with the Semantic Scholar API and the Google Scholar website to collect papers that are not benchmarks. However, such an attempt yielded an excessive volume of results, with most having a low relevance. For example, when trying to retrieve non-benchmark papers about hallucination, we obtain about 37,100 papers from Semantic Scholar, while most of them only mention "hallucination" in the abstract as background knowledge. Consequently, we refine our method and switch to examining four actively maintained GitHub repositories specializing in LLM safety literature (ThuCCSLab, 2024; Zhou, 2024; Sitawarin, 2024; Corca AI, 2024).[4] Specifically, we manually identify LLM safety-related papers from the above repositories, adhering to two main criteria: **[1]** Papers must fall within the three selected topics. **[2]** Papers are not benchmark, SOK, or survey papers. *Finally, we keep 382 screened non-benchmark papers.*

***Associated GitHub Repositories.*** We employ the Paper with Code API to search for the official code repositories corresponding to each paper. To the best of our knowledge, Paper with Code is the only systematic tool for connecting papers with code repositories. We match only repositories marked as "official," which means the repositories were provided by the authors of the corresponding papers. We then conduct manual verification to exclude any potential erroneous pairs or empty repositories. We find that the results from Paper with Code are highly accurate, with only three errors detected in 198 pairs. This may be because Paper with Code has already performed manual verification.

## 3 POTENTIAL FACTORS

Beyond the intrinsic characteristics of benchmarks, we investigate external variables that are potentially associated with their influence and code quality. We first examine the authorship dimension, as papers by renowned experts may tend to gain more attention intuitively. Similarly, we consider institutional affiliation, as papers from prestigious institutions may often receive greater recognition. Despite frequent academic exchanges, physical distance and travel policies may also be related to

---

[4]We manually verified that all collected benchmark papers were also present in the four repositories, with no additional benchmark papers identified. Thus, benchmark and non-benchmark papers' data sources can be considered similar.

Table 2: Summary of the 11 factors from five dimensions studied in this paper.

| Aspect | Factor | Description | Type |
|---|---|---|---|
| Author | Author Number | The number of the authors of a paper. | Quantitative |
| | Author Citation Count (Top-1) | The top-1 (highest) citation count among all authors in a paper. | Quantitative |
| | Author H-Index (Top-1) | The top-1 (highest) h-index among all authors in a paper. | Quantitative |
| Institution | Institution Number | The number of institutions a paper is affiliated with. | Quantitative |
| | Industry Involvement Status | Whether organizations from industry participate in a paper. | Qualitative |
| | Institution CSRankings (Top-1) | The top-1 (highest) ranking in CSRankings among all institutions a paper is affiliated with. | Quantitative |
| | Institution ARWU (Top-1) | The top-1 (highest) ranking in ARWU among all institutions a paper is affiliated with. | Quantitative |
| Geolocation | Area | The area where a paper belongs to. | Qualitative |
| | Area Number | The number of the areas to which a paper belongs. | Quantitative |
| Publication | Publication Status | The publication status of a paper. | Qualitative |
| Public Search | Search Appearance Frequency | The frequency of a paper appearing in the public search results. | Quantitative |

research dissemination. Academic concentration also varies by region—e.g., three of the four Tier 1 security conferences are always in North America. Thus, we analyze the geolocation of authors and institutions. Next, we assess publication status, as peer-reviewed papers, especially in top venues, may tend to attract more attention than preprints. Lastly, search engines play a key role in knowledge retrieval, potentially correlating a paper's influence. We thus also include public search.

Motivated by the above, we examine five dimensions (*Author*, *Institution*, *Geolocation*, *Publication Status*, and *Public Search*, details in Appendix H), encompassing eleven potential factors of both qualitative and quantitative nature. Table 2 provides a detailed summary of all 11 factors.

## 4 INFLUENCE OF BENCHMARK PAPERS (RQ1)

### 4.1 METHODOLOGY OF INFLUENCE EVALUATION

To multi-dimensionally and quantitatively reflect the influence of benchmark papers, we employ five metrics across three dimensions to quantify the papers' influence. For academic influence, we use *Citation Count* and *Citation Density* (average citations per day since public release) (Sandison, 1975; Jones et al., 2017), as prior studies (Jones et al., 2017; Sandison, 1975; Kadic et al., 2020) show they can indicate the paper influence.[5] For the influence in the open-source community, we analyze *GitHub Star Count* and *GitHub Star Density* (average GitHub stars per day since repository public release) for papers with accessible code. To measure cross-disciplinary impact, we use *Scientific Field Count*, representing the number of scientific fields (Semantic Scholar, 2023; Kinney et al., 2023) where a paper's citations appear. Citation and scientific field data come from the Semantic Scholar API, while GitHub-related data is sourced from the GitHub API. We summarize the metrics used in influence evaluation in Table 7 of Appendix P.

To understand the relative influence of benchmark papers within their respective research domains, non-benchmark papers in the same domain are selected as the control group for comparison.

### 4.2 INFLUENCE EVALUATION RESULTS OF BENCHMARK PAPERS

***Influence Evaluation.*** We mainly adopt inferential statistical analysis to evaluate the influence.[6] Specifically, we use non-parametric tests to statistically compare benchmark and non-benchmark papers across five influence-related metrics. Following rules of thumb in Statistics (Mascha & Vetter, 2018; Hogg & Tanis, 2010; VanVoorhis & Morgan, 2007), we set a minimum sample size of 25 per group, requiring us to analyze all benchmark papers collectively (see Table 1) rather than segmenting them by topic. We apply the Mann-Whitney U (M-W U) test (McKnight & Najab, 2010; Nadim, 2008) to assess statistical significance (Hayslett, 2014; Ostle, 1963; Witte & Witte, 2017) and use Cliff's delta (Hess & Kromrey, 2004; Macbeth et al., 2010; Meissel & Yao, 2024) to measure practical significance (effect size) (PSU, 2023; Peeters, 2016; Kirk, 1996; Marfo & Okyere,

---

[5]For *Citation Count/Density*, self-citations are excluded.

[6]Descriptive statistic analysis (e.g., average values) cannot be used to infer causal relationships and are sometimes misleading (Ostle, 1963; Witte & Witte, 2017; Hayslett, 2014; A. et al., 2001; Martin & Altman, 2003). Thus, we mainly adopt inferential analysis in the main text while the descriptive statistic analysis is reported in Appendix I.

2019) in distribution differences across the five metrics.[7] The methods used are robust to sample size, accommodate unequal group sizes, and require no assumptions on data distribution.

The detailed results of the M-W U test and Cliff's delta are presented in Table 3. While non-benchmark papers usually have higher average values on influence metrics (descriptive statistics in Figure 5 of Appendix I), non-parametric tests reveal that benchmark papers statistically dominate in GitHub star density and GitHub star count, with small and medium effect sizes, respectively. No significant distribution differences are found for the other three metrics. The inferential statistical analysis suggests that **benchmark papers tend to be more popular within the open-source community, though this trend may not extend to the academic community.**

Table 3: Influence M-W U test results. Positive Cliff's $\delta$: *non-benchmark* dominates; negative: *benchmark* dominates.

| Metric | $p$-value | Cliff's $\delta$ | Effect Size |
|---|---|---|---|
| GitHub Star Density | 0.012 | -0.301 | Small |
| GitHub Star Count | 0.004 | -0.347 | Medium |
| Citation Density | 0.309 | -0.112 | Negligible |
| Citation Count | 0.237 | -0.130 | Negligible |
| Scientific Field Count | 0.632 | -0.052 | Negligible |

***Correlation Between Influence Metrics.*** We analyze the monotonic correlation (also known as Spearman correlation, hereinafter simply referred to as correlation) among various influence metrics for benchmark papers, with Spearman's $\rho$ matrix shown in Figure 12 of Appendix P.[8] *Citation Density*, *Citation Count*, and *Scientific Field Count* exhibit strong correlations ($\rho$ close to 1), as all are citation-derived metrics. GitHub metrics exhibit a moderate correlation with academic influence. For instance, the $\rho$ between GitHub Star Density and Citation Density is $0.47$.

### 4.3 FACTORS IN RELATION TO THE INFLUENCE OF BENCHMARK PAPERS

We primarily use citation density to reflect the influence of a paper within the academic community, as this metric excludes the impact of time.

***Qualitative Factors.*** We study the relationships between citation density and three potential qualitative factors (industry involvement status, publication status, and area), presenting the descriptive statistics in corresponding box plots in Figure 18 of Appendix P. The descriptive statistics show that the influence seems to be associated with the factors of publication status and area. To verify it, we conduct inferential statistical analysis with a Kruskal–Wallis (K-W) test (Ostertagova et al., 2014; McKight & Najab, 2010), an extension of the M-W U test with similar properties, to assess distribution differences. However, our results show $p$ exceeding 0.05 for all three factors: industry involvement status (0.834), publication status (0.072), and area (0.152), indicating no statistically significant differences. Thus, there is insufficient statistical evidence to support that these factors correlate with citation density.

***Quantitative Factors.*** We analyze the correlation between influence and several quantitative factors from Section 3, with Spearman's (Spearman, 1904) $\rho$ matrix shown in Figure 13 of Appendix P. We use a permutation test (Phipson & Smyth, 2010) to obtain p-values and adjust them with the Bonferroni-Holm method (Holm, 1979). We observe statistical monotonic correlations between several author-prominence metrics and influence metrics. Specifically, strong positive correlations exist between *Author H-Index (Top-1)* and several influence metrics, with $\rho$ of 0.73 (Citation Count), 0.71 (Citation Density), and 0.68 (scientific field count). Additionally, *Author Citation Count (Top-1)* shows stronger correlations with *GitHub Star Count* (0.58) and *GitHub Star Density* (0.55).[9]

---

[7]Following Cliff's delta guidelines (Hess & Kromrey, 2004), effect sizes are classified as: negligible ($|\delta| \leq 0.147$), small ($0.147 < |\delta| \leq 0.330$), medium ($0.330 < |\delta| \leq 0.474$), and large ($|\delta| > 0.474$).

[8]Following Cohen's guidelines (Cohen, 1988; 1992), $\rho$ uses the coefficient mapping: weak ($0.1 \leq r < 0.3$), moderate ($0.3 \leq r < 0.5$), and strong ($r \geq 0.5$). The choice of Spearman is discussed in Appendix L.1.

[9]**Note:** We only report correlations, not causality. These correlations do not imply that author prominence must drive influence. The correlations may stem from multiple factors, such as the "Matthew Effect in Science" (Merton, 1968) (where reputation amplifies attention), intrinsic higher work quality of prominent scholars, or their interplay. Disentangling such confounding factors is currently beyond our scope and capability.

## 5 Quality of Benchmark Papers' Code Repositories (RQ2)

### 5.1 Methodology of Tool-Based Evaluation

Python is the dominant language for LLM-related libraries, and all collected repositories are primarily built on Python. Thus, we use two popular static analysis tools, Pylint (Pylint, 2024) and Radon (Radon, 2023), to assess code quality. We compute four metrics: *Pylint Score*, *Cyclomatic Complexity*, *Maintainability Index*, and *Number of Static Errors*. Also, we evaluate repository maintenance using four GitHub API-based metrics: *Reply Time*, *Last Commit Time*, *Number of Commits*, and *Commit Frequency*. These eight metrics collectively provide a view of both code quality and maintenance status (details in Appendix D). Similar to Section 4.1, we also adopt non-benchmark papers as the control group.

### 5.2 Methodology of Human-Based Evaluation

Tool-based evaluation has limitations, such as missing dynamic behavior detection and potential false negatives. To address this, we complement it with human-based evaluation.

For available repositories, we clone them and execute all code on our server (Ubuntu 20.04, four A100 GPUs). Following the README instructions, we run example scripts under recommended settings, requesting necessary access when required. If the code remains unrunnable after four hours of setup and debugging, we mark it as **not runnable**. We log any additional modifications (e.g., bug fixes) beyond those in the guides. If scripts run successfully, we record execution time. Given LLMs' high computational demands and the unpredictability of external LLM APIs, we set a four-hour time limit, exceeding that in previous studies (Collberg & Proebsting, 2016; Olszewski et al., 2023). We also manually evaluate repositories' supplementary materials, as they significantly impact usability. First, we assess the quality of install guides, which directly influence users' success and efficiency in running the code. Next, we examine the presence of data guides, which help users organize their data and execute the code effectively. Finally, we check for ethical considerations, ensuring repositories provide guidance for responsible usage, especially in LLM safety contexts where they may contain or generate harmful content. For each available code repository, we run the experimental artifact twice. Two doctoral researchers with varying levels of experience in LLMs (one junior and one senior) have conducted the two attempts independently. Our manual evaluation requires over 220 person-hours. We accept the most positive results for the repositories as the previous work (Olszewski et al., 2023). The measured metrics are summarized in Table 8 of Appendix P.

### 5.3 Tool-Based Evaluation Results of Benchmark Papers

*Code Repository Availability.* Our measurement results (Figure 10 of Appendix P) show that **benchmark papers better adhere to open science policies**. Specifically, across all topics, 87% of benchmark papers provide accessible repositories, compared to only 44% of non-benchmark papers. Examining each topic individually reveals a similar trend: for prompt injection, jailbreak, and hallucination, the proportion of benchmark papers offering available repositories reaches 100%, 75%, and 92%, respectively, significantly higher than those of corresponding non-benchmark papers.

*Code Repository Quality.* To ensure accuracy, we follow the settings of inferential analysis in Section 4.2, analyzing all benchmark papers together to meet the sample size threshold.[10] Detailed results are in Table 4. The tests show that benchmark papers dominate in Pylint score with a small effect size ($|\delta| = 0.276$), suggesting better code quality. Additionally, benchmark papers lead in three repository maintenance metrics (Reply Time, Number of Commits, and Commit Frequency), indicating that their authors are more active in maintaining and engaging with their repositories. For other metrics, no significant distribution differences are observed. In summary,

Table 4: Code quality M-W U test results. Positive $\delta$: *non-benchmark* dominates; negative: *benchmark* dominates.

| Metric | $p$-value | Cliff's $\delta$ | Effect Size |
|---|---|---|---|
| Pylint Score | 0.031 | -0.276 | Small |
| Cyclomatic Complexity | 0.649 | -0.055 | Negligible |
| Maintainability Index | 0.244 | -0.140 | Negligible |
| Number of Static Errors | 0.783 | -0.014 | Negligible |
| Reply Time (Hours) | 0.044 | -0.239 | Small |
| Last Commit Time (Days) | 0.491 | 0.083 | Negligible |
| Number of Commits | 0.001 | -0.389 | Medium |
| Commit Frequency | 0.010 | -0.309 | Small |

---

[10]The descriptive statistic analysis of code repository quality (e.g., average values) is in Appendix J.

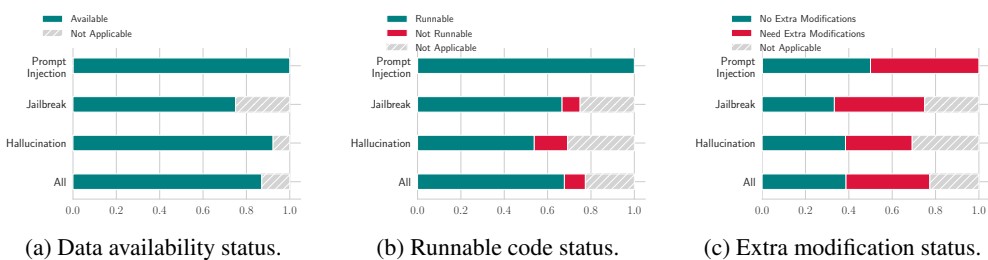

(a) Data availability status.    (b) Runnable code status.    (c) Extra modification status.

Figure 2: Human-based evaluation results of code quality.

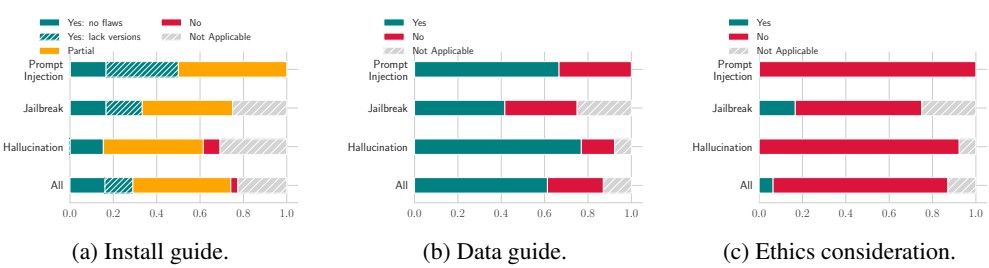

(a) Install guide.    (b) Data guide.    (c) Ethics consideration.

Figure 3: Human-based evaluation results of supplementary materials. Repositories without code or unavailable ones are labeled "Not Applicable." For install guides: repositories are categorized as "No" (no guide), "Partial" (guide with setup issues), or "Yes" (fully functional guide); if a guide includes all required Python/library/package versions, it is labeled "Yes: no flaws"; if version information is missing but the guide is still runnable, it is "Yes: lack versions."

we find that **benchmark papers exhibit higher code repository quality in terms of Pylint score and maintenance frequency**.

### 5.4 HUMAN-BASED EVALUATION RESULTS OF BENCHMARK PAPERS

***Code Repository Quality.*** We report data availability status in Figure 2a. Data is essential for LLM safety research, as running related code without it is impossible. Due to ethical or copyright concerns, some authors may only release a sub-dataset. However, if core scripts run with the provided data, we consider it available. Papers lacking public repositories or providing empty repositories are marked "Not Applicable." Our results show high data availability among all benchmark papers (0.87). For jailbreak, availability is 0.75, with all public repositories providing the necessary data. Prompt injection and hallucination reach 1.00 and 0.92, respectively.

Following the settings in Section 5.2, we assess whether benchmark papers provide runnable code. Repositories with executable scripts are labeled "Runnable," while those without are "Not Runnable." Repositories containing only data, figures, or no public code are marked "Not Applicable." Results are shown in Figure 2b. Overall, 68% of benchmark papers provide runnable scripts, while 10% contain non-executable scripts. In prompt injection, all benchmark papers offer runnable scripts. In jailbreak, the rate is 0.67. However, hallucination has a lower proportion at 0.54, with 15% of papers having non-runnable scripts. Additionally, 23% of repositories contain only data or other materials without code.

For repositories with runnable scripts, we record execution time. Even for runnable scripts, debugging and execution require a long time, averaging about 120 minutes. Since we report the most positive outcomes and our testers have relevant expertise, researchers from other fields may experience higher time costs when using these repositories. More detailed discussion and results are presented in Appendix O.

When measuring the occurrence of extra modifications needed to run example scripts, the results in Figure 2c are not ideal. **Only 39% of benchmark papers provide runnable code without any modifications, showing significant room for improvement.** This proportion remains low across safety topics: hallucination (0.38) and jailbreak (0.33), with prompt injection having the highest rate

at just 0.50. The most common issues include mismatched library/package versions and hardcoded, non-existent paths in scripts.

***Supplementary Material Quality.*** The quality results of supplementary materials are in Figure 3. Among public repositories with code, only 3% lack a useful install guide. However, **install guide quality needs improvement**—only 16% provide flawless install guides. Around 45% contain some issues, contributing to the high debugging time. Additionally, 13% provide functional guides that lack version details, with this proportion reaching 17% for jailbreak and 33% for prompt injection. As the LLM field evolves rapidly, missing version details may cause future incompatibilities, making it difficult for users to revert to compatible versions. We urge contributors to address this issue.

Most repositories publish their datasets, but they do not always include a data guide. For prompt injection, all benchmark papers provide data, yet only 67% include a corresponding guide. Overall, only 61% of benchmark papers provide a data guide. The absence of a data guide can increase usage difficulty, especially if preprocessing is required.

We further assess the inclusion of ethical considerations in repositories, as the repositories may directly help harmful response generation. Only 6% include appropriate ethical guidelines. For prompt injection, none of the repositories provides ethical considerations. While most papers discuss ethical concerns in their manuscripts, this is often absent in the associated repositories. For example, a widely-cited jailbreak benchmark repository (Chao et al., 2024) contains hundreds of successful and highly harmful jailbreak responses. Alarmingly, this code base includes no ethical considerations. These findings are especially troubling, as they underscore a significant oversight by benchmark contributors and pose a real risk of facilitating the spread of harmful content.

### 5.5 FACTORS IN RELATION TO THE CODE QUALITY OF BENCHMARK PAPERS

***Qualitative Factors.*** We primarily focus on the Pylint score, as it provides an overall measure of code quality. Box plots in Figure 19 of Appendix P illustrate its distribution with industry involvement status, publication status, and area. For industry involvement and publication status, distributions show no clear differences, with similar mean and median values across groups. The K-W test yields $p > 0.950$, far above 0.05, confirming no significant differences. For the factor area, North America has the widest range of Pylint scores, while Europe has the most concentrated distribution, with the highest average and median scores. However, this does not indicate a significant correlation, as their K-W test gives $p = 0.274$, exceeding 0.05. In conclusion, insufficient evidence supports industry involvement status, publication status, or area as factors correlating with Pylint scores.

***Quantitative Factors.*** We analyze the correlation between code quality metrics and quantitative factors from Section 3 (Spearman $\rho$ matrix in Figure 15 of Appendix P). While high-influence authors are often associated with popular benchmarks ( Section 4.3), their involvement **is not necessarily associated with** higher code quality. **The author's h-index (top-1) and citation count (top-1) show no significant correlation with code quality indicators**($p_{\text{adjusted}} > 0.05$). However, we observe a strong negative correlation between ARWU ranking and code maintainability index ($\rho = -0.57$). These suggest that higher-ranked institutions are associated with more maintainable code.

## 6 RELATIONSHIPS BETWEEN BENCHMARK PAPERS' INFLUENCE AND CODE REPOSITORY QUALITY (RQ3)

We examine the influence of **benchmark papers**, particularly their academic influence, in relation to code quality. As in Section 4.3, we use citation density as the primary metric to measure influence.

***Qualitative Factors.*** We analyze the relationship between citation density and the availability of code repositories and datasets (Figure 16 of Appendix P). Papers with available code and data show higher average and median citation density, but the correlation is not statistically significant (K-W test, $p = 0.058$). Next, we examine citation density in relation to extra modifications and runnable code (Figure 17 of Appendix P). The K-W test yields $p$ of 0.007 and 0.002, indicating significant distribution differences in at least one group. We then perform Dunn's test (Pohlert, 2014; Dinno, 2015) as a post-hoc analysis for finer-grained insights. When examining whether code is runnable, we find that papers providing runnable code exhibit a significantly higher citation density compared to those without accessible code ($p = 0.004$). Furthermore, **when considering the extent**

**of code modification required, compared with papers without accessible code, those offering code that can be used without additional modification show significantly higher citation density ($p = 0.005$), while code requiring modification does not yield a significant difference .** For other pairs, no conclusions could be drawn ($p > 0.05$).

***Quantitative Factors.*** We compute Spearman $\rho$ between code quality metrics (including tool-based and repository maintenance metrics) and influence metrics (Figure 14 of Appendix P). None of the influence metrics are observed to have a significant correlation with code quality indicators ($p_{\text{adjusted}} > 0.05$), suggesting that **code following a higher coding standard is not necessarily linked to more citations.**

In summary, our findings show that a paper's popularity is positively correlated with the availability of **runnable** code. Conversely, factors such as low tool-based evaluation scores and infrequent maintenance do not appear to be significant impediments to a paper's widespread adoption. The academic community may prefer "pragmatism": researchers value functional code in benchmark papers but do not strongly consider the coding standards, quality, and maintenance.

## 7 ISSUES AND ADVICE

***Non-User-Friendly Code Repositories.*** While many code repositories are functional, they are not user-friendly. Users have to debug and modify multiple files to run the code. We observe that the most common issues are inconsistent styles and input handling, which likely stem from benchmarks reusing code from various repositories. For example, API keys may be managed in different ways, and hardcoded absolute paths frequently cause execution failures. While fully refactoring imported code is impractical, **contributors should wrap reused segments to ensure a uniform interface and use relative paths for better user-friendliness, portability, and stability.**

***Flawed Supplementary Materials.*** First, many repositories have flawed install guides, often lacking clear Python versions or library dependencies. Compatibility issues may arise quickly, as seen with `openai-python`, which underwent major changes in `v1.0.0`, causing runtime errors (OpenAI, 2024b;a). Poorly maintained environment files (e.g., `requirements.txt`) further complicate setup, often containing unnecessary dependencies or local paths. Data-related issues are also common, such as missing usage instructions or mismatches between provided data and code expectations. These problems hinder effective usage. **Contributors should minimize dependencies, create clean environment files, and clarify data usage to enhance repository usability.** Second, LLM safety benchmark repositories may help harmful content generation, yet many lack ethical warnings. Some even include harmful jailbroken responses without supervision or user warnings. **Authors should integrate ethical guidelines within repositories as they do in the papers.**

## 8 DISCUSSION AND LIMITATION

### 8.1 DISCUSSION

***Open Challenges.*** Some of the benchmark deficiencies mentioned may be inherently unsolvable. The LLM field evolves quickly, with frequent updates to popular libraries like `vLLM` (vLLM, 2025), leading to structural changes and compatibility issues. Maintaining benchmark repositories requires continuous tracking, but this is difficult due to short-term academic funding and contributors' shifting priorities. Unlike academia, open-source communities may provide better long-term support, ensuring sustainable maintenance and high-quality code.

***Community Perspectives.*** To understand how the LLM safety community views the research questions addressed in this study and to understand whether the issues we identify align with community expectations and concerns, we conducted a 17-question anonymous survey distributed through public academic channels (details in Appendix M and Appendix N). Respondents broadly endorsed our metric choices, agreed that author, institution, and geolocation affect influence, and highlighted code usability as a core point: most expected at least runnable code with a minimal example, and considered installation and data guides essential. These community perspectives corroborate the issues observed in our empirical analysis and highlight their broader relevance to real-world research.

## 8.2 LIMITATION

***Potential Collection Biases.*** As detailed in Section 2.2, we collect papers using automated tools with manual verification, yet some papers or repositories may still be inadvertently missed. To reduce this risk, we verified that all benchmark papers in our dataset also appear in the four major community repositories, and our checks did not reveal any benchmark or non-benchmark code hosted outside GitHub. Although Papers with Code provides extensive human-validated mappings, a small number of mislinked or unreported repositories may remain; however, their rarity suggests minimal effect.

***Human-Based Evaluation Biases.*** Human-run reproducibility tests introduce additional sources of variability. Different evaluators may encounter different debugging paths or spend disproportionate time on minor issues. To mitigate this, we avoid binary outcomes whenever possible and require detailed justification for all manual assessments. When evaluators obtain divergent results, we adopt the more positive outcome to reduce the known bias toward reporting overly negative findings (Olszewski et al., 2023; Collberg & Proebsting, 2016). While these steps reduce inconsistency, they cannot fully eliminate subjectivity in human-based evaluation.

***Scientific Quality Control.*** Our study cannot fully control for the intrinsic scientific quality of benchmark papers. All benchmark papers originate from arXiv or comparable platforms and thus pass a basic screening threshold, naturally removing the lowest-quality manuscripts. However, distinguishing exceptionally high-quality benchmarks lies beyond current methodological capabilities. More fundamentally, meta-research lacks a widely accepted, scalable measure of intrinsic scientific or dataset quality; existing proxies (e.g., length-based indicators or rare experimental designs) are coarse and limited (Xie et al., 2019; Falagas et al., 2013; Mammola et al., 2022; Larivière & Gingras, 2010). Additionally, disentangling potential confounding factors, such as the influence of author prominence versus the intrinsic quality of the benchmark, is currently beyond our scope and capability. We therefore rely on observable code-level proxies (static analysis metrics, reproducibility tests), which do not fully capture scientific quality. Developing reliable, domain-agnostic scientific-quality measures remains an open challenge.

***Imperfect Metrics.*** Finally, several metrics used in our analysis are inherently imperfect indicators of the underlying constructs they aim to measure. For example, citation-related metrics capture only partial aspects of scientific influence (Maiti et al., 2023; Bornmann & Daniel, 2007; da Silva & Memon, 2017; Gregor et al., 2023; Sharma & Spinellis, 2020). As is standard in meta-research, we rely on such imperfect but practically measurable proxies, and we mitigate their limitations by employing multiple complementary metrics rather than depending on any single indicator. Nevertheless, no combination of proxies can fully capture the constructs of interest, consistent with longstanding observations that no perfect scientometric indicator exists (Ravenscroft et al., 2017; Agarwal et al., 2016; Fire & Guestrin, 2019).

## 9 CONCLUSION

We conduct a multi-dimensional evaluation of LLM safety benchmark papers' influence and code quality, analyzing factors affecting both and their correlation. Our findings show that LLM safety benchmarks have less academic influence than expected, with no significant difference from non-benchmark papers. We observe statistical associations between certain author-prominence measures and influence metrics. However, we do not detect statistically significant correlations between those author-prominence measures and code-quality indicators. Benchmark papers still have significant room for improvement in both code quality and supplementary materials. Issues such as flawed installation guides and problematic code are still widespread and continue to hinder usability. Ready-to-use code positively relates to a paper's influence, but other code quality standards show no significant correlation. Finally, we provide targeted advice to improve benchmarks' repository quality, aiming to better support future research.

ETHICS STATEMENT

This study complies with the ICLR Code of Ethics and established norms for responsible meta-research. All data analyzed in this work are publicly available and non-sensitive, including citation metadata, author affiliations, and open-source repository information. No proprietary datasets, private communications, or non-public artifacts were accessed at any stage. We do not process personally identifiable information beyond what is already openly disclosed by the authors themselves in academic publications and public code repositories. Harmful or unsafe content, such as jailbreak examples or security-relevant artifacts, is explicitly excluded from our dataset. All analyses are conducted at an aggregated level and do not target, profile, or evaluate individual researchers.

Our survey component was conducted under strict ethical and privacy safeguards. Participation was entirely voluntary. Respondents could choose, at their own discretion, whether to disclose background quasi-identifiers such as their geographic region or academic training. No personally identifiable information (PII) was collected for participation. The survey was distributed through public academic channels without revealing the focus or hypotheses of this manuscript. Responses were retained only in anonymized form and analyzed exclusively at the group level. The survey did not involve any interventions, personal data collection, or sensitive information, and therefore does not constitute human-subjects research requiring IRB approval under standard institutional guidelines.

REPRODUCIBILITY STATEMENT

Our measurement results are based on public APIs and repositories, and thus, there is no barrier to reproduction. The data collection code is available in our anonymous repository. However, on July 24, 2025, the platform Papers With Code suddenly discontinued (HyperAI, 2025), and the data from it is unavailable anymore. Considering this, we promise that, upon acceptance of the manuscript, the organized and cleaned datasets (such as the retrieved metadata and the human evaluation logs) used in the paper will be made available upon request.

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

## A  LLM USAGE

We acknowledge the use of LLM in this manuscript. We used LLM to polish the manuscript, correct grammatical errors, and resolve some issues during the LaTeX compilation. LLMs are also used to help anonymize and clean up our code repository.

## B  INTRODUCTION TO STUDY SCOPE

Although LLMs have shown their strong capability, they are facing safety risks, as they are susceptible to a variety of sophisticated attacks, such as backdoor attacks (Bagdasaryan & Shmatikov, 2022; Chen et al., 2021), data extraction attacks (Carlini et al., 2021; Lukas et al., 2023), and memorization-related attacks (Mireshghallah et al., 2022; Tramèr et al., 2022; Chu et al., 2024; Wen et al., 2024).

Among various attacks, there are several attacks specific to LLMs. Prompt injection attacks are one of the most notable ones. In such attacks (Perez & Ribeiro, 2022; Greshake et al., 2023), the adversary could mislead the target LLM by adding designed texts to the original queries. Jailbreak attacks are another popular attack type. The adversary conducts jailbreak attacks (Liu et al., 2023; Deng et al., 2023; Wei et al., 2023; Li et al., 2023a; Shen et al., 2023; Wang et al., 2023; Chu et al., 2025b; Jiang et al., 2025) by using different methods to bypass the safeguards of LLMs and induce LLMs to complete the given harmful tasks. Hallucination (Rawte et al., 2023; Ji et al., 2023; Li et al., 2023b) is another inherent safety risk of LLMs, where LLMs generate content that appears correct but actually contains errors or contradicts facts.

These three risks could cause very serious consequences in the real world. For example, an attacker could use prompt injection to mislead an LLM into performing unintended, potentially dangerous actions. Jailbreak techniques could enable LLMs to execute forbidden, hazardous tasks. Hallucination, particularly in specialized fields such as medicine, poses severe risks: incorrect responses could lead to wrong decisions of human beings, sometimes even endangering lives. On the other hand, these three topics represent safety threats unique to LLMs, potentially embodying distinctive characteristics within the LLM safety landscape.

Therefore, in this study, we focus on these three newly emerged safety topics related to LLMs, including **prompt injection**, **jailbreak**, and **hallucination**. Our study aims to uncover valuable insights specific to the unique challenges in LLM safety.

## C  DETAILS OF KEYWORDS

***Search Results.*** The keyword set for search results is:

- `llm+OR+large+language+model+`
  `[Safety Topic]+`
  `assessment+OR+evaluation+OR+benchmark`

`[Safety Topic]` is one of the following: prompt injection, jailbreak, and hallucination.

***Collection of Benchmark Papers.*** The keyword sets to collect benchmark papers are:

- `large language model`
  `[Safety Topic] benchmark`

- `large language model`
  `[Safety Topic] evaluation`

- `large language model`
  `[Safety Topic] assessment`

`[Safety Topic]` is one of the following: prompt injection, jailbreak, and hallucination.

## D    DETAILS OF CODE QUALITY METRICS

- **Pylint Score** It is a global evaluation score for codes, with a maximum score of 10.0. It is computed based on multiple aspects, such as the percentage of errors and warnings for each module. The higher the score, the better the quality.

- **Cyclomatic Complexity** It is a measurement to determine the stability and level of confidence in a program (McCabe, 1976). It measures the number of linearly independent paths through a program module. Programs with lower cyclomatic complexity are easier to understand and less risky to modify.

- **Maintainability Index** It is a software metric that measures how maintainable (easy to support and change) the source code is. The higher the score, the easier the code is to maintain. We follow the settings in Microsoft Visual Studio (Microsoft, 2022), which uses a shifted scale (0 to 100) derivative.

- **Number of Static Errors** Errors that occur during compile time are called static errors. Here we count the total number of static errors in the entire code.

- **Reply Time** It measures the average time between when an issue is raised in a repository and when it first receives a response from a contributor.

- **Last Commit Time** It measures the time interval between the last commitment in the repository and the current date.

- **Number of Commits** It measures the total number of commits since the repository is created.

- **Commit Frequency** It measures the commit frequency of a code repository by dividing the total number of commits by the repository's existence duration.

## E    DETAILS OF DATA SOURCES

***Semantic Scholar (Semantic Scholar, 2024) and Google Scholar (Google, 2024).*** Semantic Scholar includes over 200 million publications from all fields of science and comprehensively covers papers from various publishers and preprint databases, making it one of the most extensive academic databases available. In this study, we utilize the Semantic Scholar API to gather metadata on a range of papers and the corresponding authors, including attributes such as citation count of papers, publication venue of papers, citation count of the authors, etc. Note that previous studies (Hannousse, 2021) have shown that for computer science, Semantic Scholar and Google Scholar have similar index coverage, with each missing only a small number of articles. As such, we also use Google Scholar to complement our data collection. Since Google Scholar lacks an official API, we mainly use the Semantic Scholar API for data retrieval. We only manually complement missing data from Google Scholar when it is unavailable in Semantic Scholar.

***Paper with Code (Meta AI, 2024).*** Finding the corresponding official code repositories for papers is a challenge because papers usually do not introduce or publish their corresponding official code repositories in a unified manner. Papers with Code, maintained by Meta, addresses this by centrally linking papers to their code, datasets, and evaluation results, promoting reproducible research in computer science. To the best of our knowledge, the Paper with Code API is the only tool that allows us to retrieve the connections between academic papers and code repositories.

***GitHub (GitHub, 2024).*** GitHub enables collaboration and code sharing on a global scale. It is the world's largest source code host as of June 2023.[11] For this study, we leverage the GitHub API to systematically collect metadata on code repositories linked to various research papers, enabling us to analyze and integrate coding resources directly associated with academic publications. From this API, we collect metadata of code repositories, such as GitHub star count, number of commits, reply time of issues, etc. We discuss the issues about other code hosting platforms in Ethics Statement.

---

[11] https://en.wikipedia.org/wiki/GitHub

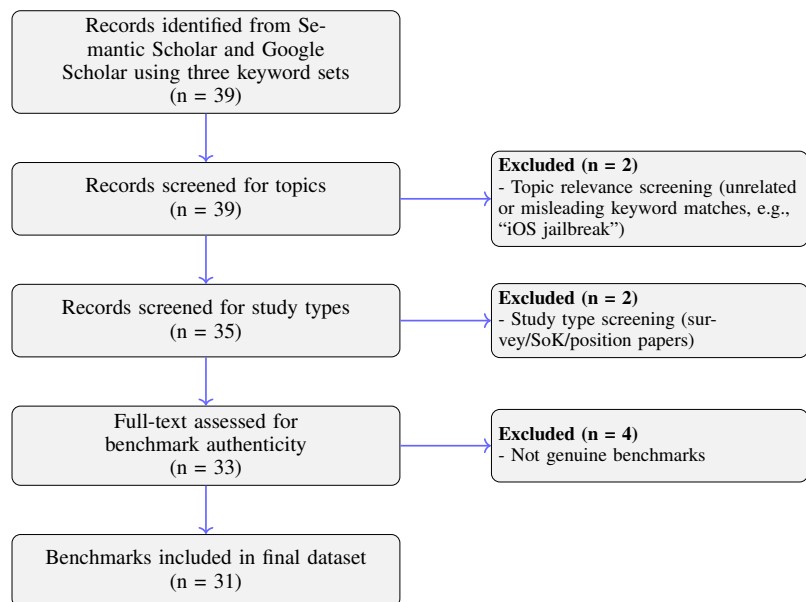

Figure 4: PRISMA-style flow diagram for benchmark selection.

## F PRISMA Flow Diagram of Benchmark Paper Selection

We apply a three-stage manual screening procedure to ensure that the collected benchmarks are both relevant and methodologically appropriate for our study:

**(1) Topic relevance screening.** Papers must genuinely concern LLM safety and address one of the selected safety topics, rather than merely matching keywords in unrelated domains (e.g., "iOS jailbreak").

**(2) Study type screening.** We exclude surveys, SoK papers, position papers, and commentaries, and retain only primary research papers.

**(3) Benchmark authenticity screening.** Papers must present a genuine benchmark (i.e., defining benchmark tasks, datasets, or evaluation protocols). Works that merely use existing benchmark datasets for evaluation (e.g., "we evaluate our method on benchmark datasets") are excluded.

We present the entire PRISMA (PRISMA Executive, 2020) diagram for benchmark selection in Figure 4.

## G Details of Metadata

For each paper, we collect the following metadata, including both the paper's metadata and its code repository's metadata (if the repository is available):

**From Semantic Scholar:** paper title, author list, scientific field list, paper release date, paper's citation count, each author's citation count.

**From GitHub API:** repository release date, repository's GitHub star count, number of commits, issue/pull request thread (including the time stamps and the corresponding users).

## H Introduction to the Factor Dimensions

In this paper, we investigate the five dimensions (*Author*, *Institution*, *Geolocation*, *Publication Status*, and *Public Search*), covering eleven potential factors, including both qualitative and quantitative ones. We summarize the details of eleven factors in Table 2.

***Author.*** We employ *Author Number*, *Author Citation Count (Top-1)*, and *Author H-Index (Top-1)* as the candidate factors. Note that we use the Semantic Scholar API to retrieve related data, and we only consider the top-1 citation count and top-1 h-index among all the authors. The top-1 citation count and h-index may come from different authors of a given paper.

***Institution.*** We manually retrieved benchmark papers' PDFs on Semantic Scholar to compile a list of institutions with which benchmark articles are affiliated. We attempt to use two popular rankings, including Academic Ranking of World Universities (ARWU)[12] and CSRankings[13], to quantify the reputation of the institutions separately. We only consider the top-1 rankings in ARWU 2024 and CSRankings (2014–2024) among all the involved institutions of a given paper. Additionally, we consider the type of institution, studying whether industry involvement positively contributes to the influence of benchmark papers. In summary, we study *Institution Number*, *Insitution ARWU (Top-1)*, *Insitution CSRankings (Top-1)*, and *Industry Involvement Status* in this dimension.

***Geolocation.*** In the dimension of geolocation, we investigate two factors: *Area* and *Area Number*. In this paper, we study the impact of areas by continent, and Asia and Oceania are considered as a whole as the Asia-Pacific region.

***Publication Status.*** We take the *Publication Status* into consideration. The computer science domain places more emphasis on conferences. Therefore, we use the taxonomy from the CSRankings. We denote papers published in CSRankings recommended conferences as "published-leading," papers published in other conferences as "published-other," and papers published in workshops or preprints as "unpublished." To ensure accuracy, we manually searched the publication status of each benchmark paper on Semantic Scholar (on November 1, 2024).

***Public Search.*** We study the potential impact of the frequency of benchmark papers appearing in the public search results (referred to as *Search Appearance Frequency*) on their influence. To avoid the cookie effects and fetch stateless results, we use the Google Custom Search API and its default settings. Specifically, for each safety topic, we use the sets of keywords in Appendix C to obtain the first 50 search results, and then count the appearance frequency of each paper.

## I DESCRIPTIVE STATISTICAL ANALYSIS OF INFLUENCE EVALUATION

The average values of five influence-related metrics are presented in Figure 5. When considering all papers, we find that the average values of benchmark papers lag behind those of non-benchmark papers across all five influence-related metrics. Detailedly, on the metrics related to the academic community (i.e., citation count and citation density), non-benchmark papers have a higher average citation count and density than benchmark papers (47.838 vs. 30.600, 0.114 vs. 0.095). In terms of the number of scientific research fields affected by papers, both are close, but non-benchmark papers maintain a slight advantage, with a lead of 0.131 on the average values. We have similar observations on the metrics related to the open-source community. Non-benchmark papers have, on average, 54.574 more GitHub stars than benchmark papers and lead in the metric star density by 0.055.

For each safety topic, results vary slightly. In prompt injection, benchmark papers show higher means of citation density and scientific field count than non-benchmark papers (0.075 vs. 0.046, 3.406 vs. 2.303) but have similar citation count means (20.000 vs. 19.727). For jailbreak papers, benchmark papers outperform non-benchmark papers in terms of citation density, GitHub star count, and GitHub star density. However, in hallucination papers, benchmark papers lag behind non-benchmark papers in the means of all metrics except scientific field count, with mean values below 60% of non-benchmark papers for most metrics.

In our previous measurements, we find that both non-benchmark and benchmark papers impact an average of over four scientific fields, which indicates that LLM safety papers also hold influence in other domains. We conduct an in-depth analysis of this aspect, and the distribution of scientific fields is illustrated in Figure 6. Considering Computer Science, Linguistics, and Engineering as fields directly related to LLMs, and all others as indirectly related, we find that benchmark papers impact 17 indirectly related fields, while non-benchmark papers affect an additional three indirectly related

---

[12]https://www.shanghairanking.com/news/arwu/2024
[13]https://csrankings.org/

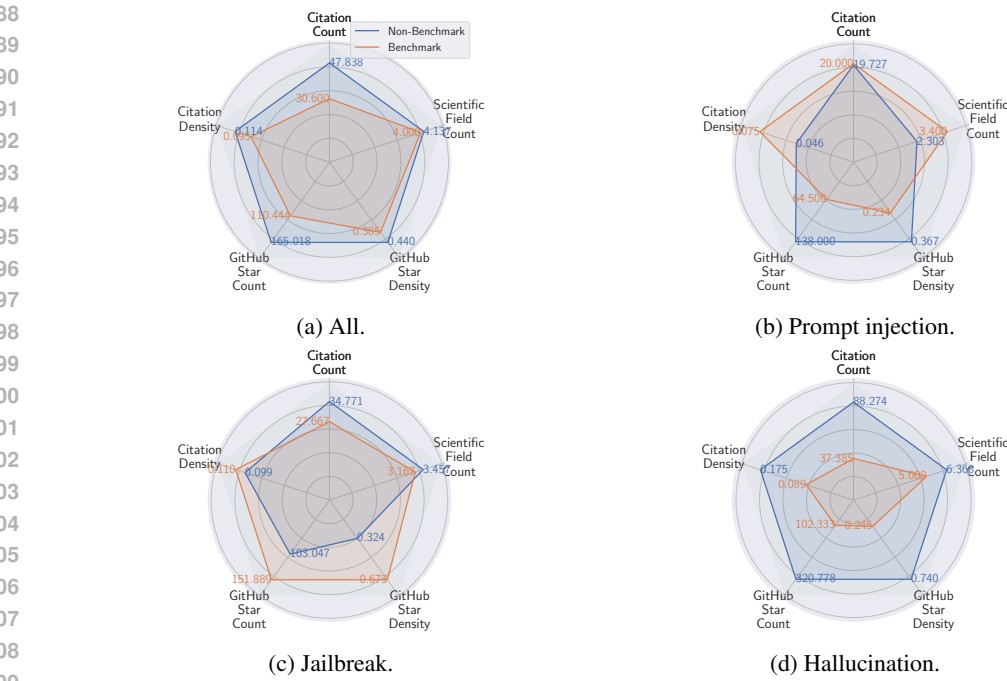

Figure 5: Average values of five influence-related metrics on benchmark and non-benchmark papers.

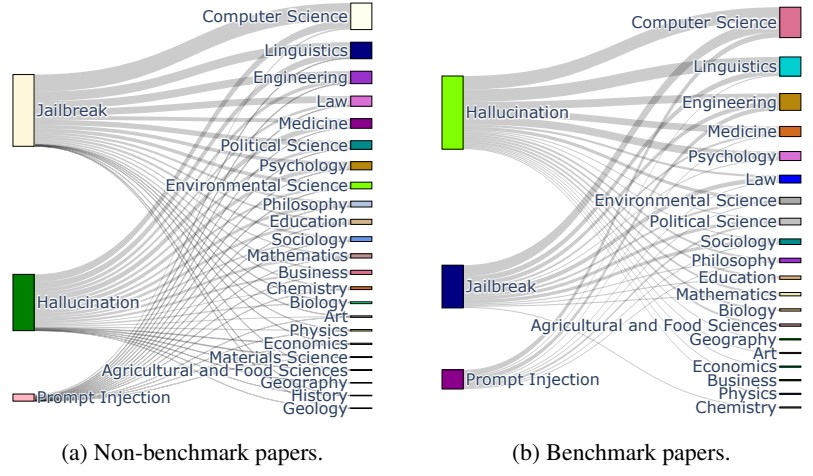

(a) Non-benchmark papers.                    (b) Benchmark papers.

Figure 6: Distribution of the scientific fields that the LLM safety papers influence.

fields beyond this. For non-benchmark papers, the most impacted indirectly related field is Law, with 110 such associations. For benchmark papers, the leading indirectly related field is Medicine, with nine associations. We also observe that LLM safety papers impact some fields beyond our expected range, such as Arts and Materials Science. The above results indicate that LLM safety papers have considerable cross-field influence, highlighting the importance of LLM safety research.

We explore the relationship between the cumulative distribution of benchmark papers' citation counts and GitHub star counts over time. Identifying general patterns is challenging due to variations in the timing of code and paper releases—codes may be released before, after, or simultaneously with the paper. The consistent pattern observed is that when code is released before the paper, the repository's GitHub star count surges upon the paper's publication. Of 12 benchmark papers with this release timing, 11 follow this trend (Figure 11 of Appendix P). The exception likely reflects the paper's low citation count (only one), limiting community interest in its paper and code.

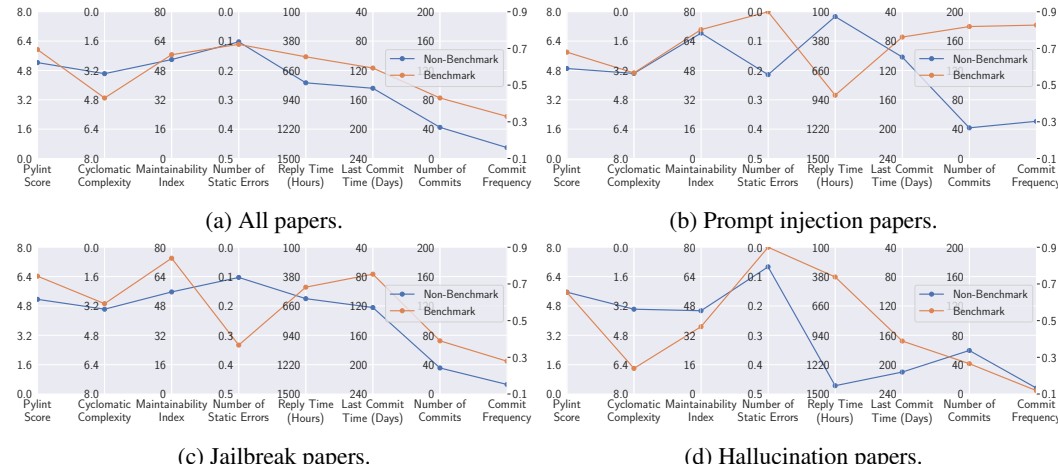

(a) All papers.

(b) Prompt injection papers.

(c) Jailbreak papers.

(d) Hallucination papers.

Figure 7: Average values of eight metrics related to the code repository quality on benchmark and non-benchmark papers. We have flipped the axes as needed, so now all points/lines on the upper side indicate better quality or performance.

## J    DESCRIPTIVE STATISTICAL ANALYSIS OF TOOL-BASED EVALUATION

***Code Repository Quality.*** The descriptive results of tool-based code quality evaluation are presented in Figure 7. When considering benchmark papers across all safety topics, we observe that their associated repositories lead or are close to non-benchmark papers' in the mean values of most metrics, with average cyclomatic complexity being the only exception. Specifically, benchmark papers outperform non-benchmark papers in both average Pylint scores and average maintainability index (5.937 vs. 5.229, 56.606 vs. 53.940). On the average number of static errors, benchmark papers are quite close to non-benchmark papers (0.111 vs. 0.102). From the perspective of mean values, benchmark papers outperform non-benchmark papers in all metrics related to code maintenance frequency. For instance, the average reply time for benchmark papers is significantly shorter than for non-benchmark papers, reduced by 254.204 hours; additionally, the average maintenance frequency of benchmark papers is nearly double that of non-benchmark papers (0.329 vs. 0.159).

A granular analysis by safety topic reveals slightly different observations. For prompt injection, benchmark papers generally outperform or match non-benchmark papers in the means of code quality metrics, with only a slight lag in cyclomatic complexity means (0.026). However, their average reply time is significantly longer than non-benchmark ones by 750.683 hours. In jailbreak, benchmark papers excel in nearly all metric means, notably in the maintainability index (74.032 vs. 55.647). The exception is the average number of static errors, where benchmark papers have higher (0.333 vs. 0.103), though the absolute value remains low ($< 1$), indicating room for improvement. For hallucination, benchmark papers show similar Pylint average scores but fare worse in average cyclomatic complexity ($\uparrow$3.377) and average maintainability index ($\downarrow$8.699). Notably, the average reply time for benchmarks is over 1,000 hours shorter than for non-benchmarks.

## K    RELATED WORKS

***Meta-Study.*** Over the past decades, researchers have conducted numerous meta-studies on academic papers, journals, and conferences. The Matthew effect—where prominent authors gain more visibility—is identified as early as 1968 (Merton, 1968). Later studies examine factors influencing paper impact, such as publication venue and paper length (Wang et al., 2024; Callaham et al., 2002; Teplitskiy et al., 2022; da Silva, 2021; Velez-Estevez et al., 2022; Kadic et al., 2020), assess the influence measurement, and classify citations semantically (Kunnath et al., 2021; Vaioulis et al., 2015). Other studies explore the evolution of publication venues, including changes in author guidelines (Malički et al., 2021; Alder et al., 2021; Garfield, 1972). In computer science, meta-studies begin in 1979 with an article studying paper distribution within the domain (Salton & Bergmark,

1979), and recent studies reveal rapid publication growth and category clustering (Devarakonda et al., 2020).

***Code Quality Evaluation.*** Some previous studies (Trisovic et al., 2022) focus on public code-hosting databases like the Harvard Dataverse repository. Some other studies in computer science evaluate the availability of artifacts in different venues (National Academies of Sciences, Engineering, and Medicine, 2019; Vandewalle et al., 2009; Raghupathi et al., 2022; Collberg & Proebsting, 2016), such as various ACM and IEEE journals and conferences. In addition, some researchers (Raff, 2019; Hamm et al., 2019; Dacrema et al., 2019; Gundersen et al., 2019) dive deep into the ML and AI domains specifically, as such domains are more affected by randomness. For instance, previous research (Gundersen et al., 2019) shows that publications at AAAI conferences currently fall short of providing enough documentation to facilitate reproducibility. In the security field, there are also studies measuring code quality (Olszewski et al., 2023; Hamm et al., 2019; van der Kouwe et al., 2019). A study (van der Kouwe et al., 2019) examines 50 system security papers and points out that one-fourth of them did not clearly provide software or platform versions. Another work (Olszewski et al., 2023) assesses the availability and reproducibility of code for over 700 security papers related to machine learning.

## L  EXPLORATORY MULTIPLE LINEAR REGRESSION ANALYSIS

### L.1  SELECTION OF CORRELATION ANALYSIS METHOD

The total number of samples in the Benchmark dataset is relatively limited. In our study, we included all relevant benchmark works that we retrieved, with a total size of 31.

According to statistical guidelines, approximately ten observations per predictor are needed to obtain stable estimates in regression (Statsols, 2024; Pripp, 2024; Newsom, 2021). With eight independent variables, a size of 31 provides insufficient degrees of freedom for a robust multivariate analysis. A Multiple Linear Regression model under these conditions would likely suffer from multicollinearity and unstable parameter estimates.

Instead, the correlation coefficient requires a relatively small sample size (according to statistical guidance, a sample size of 25 or higher is sufficient) (Minitab Support, 2024; McIntyre & David, 1938). Therefore, we prioritized correlation coefficients to explore the strength and direction of bivariate associations, which offers a more reliable interpretation given the limited data availability.

Furthermore, we selected Spearman's rank correlation coefficient $\rho$ rather than Pearson's $r$ to mitigate issues related to the distribution of the data. Compared with Pearson, Spearman has lower data requirements, and its $\rho$ offers greater stability against outliers, which can disproportionately influence results in limited sample sizes (Spearman, 1904; Phipson & Smyth, 2010).

We assessed statistical significance using permutation tests rather than standard asymptotic methods, which are often ill-suited for small datasets. By performing 10,000 random shuffles, the permutation test yields exact empirical $p$-values rather than approximations. This ensures robust inference despite the limited number of observations and the potential non-normality of the underlying population (Good, 2005; Edgington & Onghena, 2007; Ernst, 2004).

Finally, to account for the risk of Type I errors (false positives) arising from repeated testing, we applied the Bonferroni-Holm correction (Holm, 1979). This step strictly controls the Family-Wise Error Rate (FWER). To maintain rigorous standards, we only report $\rho$ results with adjusted $p$-values $< 0.05$ as statistically significant in the main text. Acknowledging that strict FWER control increases the risk of Type II errors (false negatives), we also report $\rho$ results of the raw (unadjusted) $p$-values in the Appendix. These unadjusted results should be interpreted as exploratory signals requiring further validation in larger cohorts.

### L.2  EXPLORATORY MULTIPLE LINEAR REGRESSION

Below, we additionally report two exploratory multiple linear regression models based on ordinary least squares (OLS) fitted on the benchmark dataset (N = 31). Before performing linear regression, we standardized all predictor variables. Given the sample size (N = 31) and the relatively large number of predictors (eight per model), the results should be interpreted with caution. The following

regressions are statistically underpowered and numerically unstable. We therefore include them only as supplementary evidence.

### L.2.1 REGRESSION ON AUTHOR AND INSTITUTION FACTORS

**Dependent variable:** Citation Count
**Predictors:** Author Number, Institution Number, Area Number, Author H-Index (Top-1), Author Citation Count (Top-1), Institution CSRankings (Top-1), Institution ARWU (Top-1), Search Appearance Frequency

Table 5: OLS Regression Results: Author & Institution Factors

| Predictor | Coef. | Std. Err. | t-value | p-value |
|---|---|---|---|---|
| Intercept | -2.97e-17 | 0.143 | -0.0002 | 1.000 |
| Author Number | -0.0802 | 0.172 | -0.466 | 0.646 |
| Institution Number | 0.5197 | 0.178 | 2.919 | 0.008 |
| Area Number | -0.3347 | 0.182 | -1.839 | 0.079 |
| Author H-Index (Top-1) | 0.9827 | 0.285 | 3.452 | 0.002 |
| Author Citation Count (Top-1) | -0.7993 | 0.284 | -2.815 | 0.010 |
| Institution CSRankings (Top-1) | -0.0246 | 0.175 | -0.141 | 0.889 |
| Institution ARWU (Top-1) | -0.0306 | 0.180 | -0.170 | 0.867 |
| Search Appearance Frequency | 0.1658 | 0.169 | 0.978 | 0.339 |

*Model statistics:*
$R^2 = 0.549$,   Adjusted $R^2 = 0.384$
F-statistic = 3.342,   p = 0.0118
Omnibus p = 0.003,   JB p = 0.0038,   DW = 2.481

**Notes.** The model (in Table 5) explains approximately 55% of the variance ($R^2 = 0.549$), but the adjusted $R^2$ drops to 0.384 due to the number of predictors relative to the small sample size. Several coefficients (Author H-Index (Top-1), Institution Number, Author Citation Count (Top-1)) reach nominal significance. However, the negative coefficient on Author Citation Count (Top-1) is likely caused by multicollinearity and small-N instability. These results should be treated as exploratory.

### L.2.2 REGRESSION ON CODE QUALITY FACTORS

**Dependent variable:** Citation Count
**Predictors:** Pylint Score, Cyclomatic Complexity, Maintainability Index, Number of Static Errors, Server Reply Time (Hours), Last Commit Time (Days), Number of Commits, Commit Frequency

Table 6: OLS Regression Results: Code Quality Factors

| Predictor | Coef. | Std. Err. | t-value | p-value |
|---|---|---|---|---|
| Intercept | -2.97e-17 | 0.159 | -0.0002 | 1.000 |
| Pylint Score | -0.1897 | 0.182 | -1.040 | 0.310 |
| Cyclomatic Complexity | 0.3465 | 0.257 | 1.350 | 0.191 |
| Maintainability Index | -0.0240 | 0.178 | -0.135 | 0.894 |
| Number of Static Errors | 0.4453 | 0.165 | 2.699 | 0.013 |
| Server Reply Time (Hours) | 0.1809 | 0.167 | 1.081 | 0.291 |
| Last Commit Time (Days) | 0.0045 | 0.291 | 0.015 | 0.988 |
| Number of Commits | 0.9264 | 0.635 | 1.459 | 0.159 |
| Commit Frequency | -0.9875 | 0.626 | -1.579 | 0.129 |

*Model statistics:*
$R^2 = 0.444$,   Adjusted $R^2 = 0.242$
F-statistic = 2.195,   p = 0.0690
Omnibus p = 0.019,   JB p = 0.0445,   DW = 2.240

**Notes.** The overall regression (in Table 6) does not reach the 0.05 significance threshold (p = 0.069). The adjusted $R^2$ is only 0.242. Apart from Number of Static Errors, none of the code-quality metrics exhibits a stable linear association with citation counts under this small-N setting. These results should therefore be considered exploratory.

### L.3 SUMMARY

Across both models, the limited sample size (31 observations) combined with eight predictors per model results in high-variance, low-power estimates and unstable coefficients. These regressions are included for completeness but are not used to support substantive claims.

## M LLM SAFETY BENCHMARK SURVEY

SECTION A — PAPER AND REPOSITORY INFLUENCE

**Q1.** In your opinion, which metric is more appropriate for measuring a paper's influence?

- Citation count
- Citation density
- Other: _______________________________

**Q2.** In your opinion, which metric is more appropriate for measuring a code repository's influence?

- GitHub star count
- GitHub star density
- Other: _______________________________

**Q3.** Do you think an author's prominence affects the influence of a paper or its associated repository?

- It affects papers only
- It affects repositories only
- It affects both
- It affects neither

**Q4.** Do you think the institution behind a work affects the influence of its paper or repository?

- It affects papers only
- It affects repositories only
- It affects both
- It affects neither

**Q5.** Do you think geographical location (e.g., North America / Europe / Asia-Pacific) affects the influence of a paper or repository?

- It affects papers only
- It affects repositories only
- It affects both
- It affects neither

**Q6.** Do you think publication status (e.g., top conference, other conference, preprint) affects a paper's influence?

- It affects papers only
- It affects repositories only
- It affects both
- It affects neither

**Q7.** Do you think public search visibility (e.g., Google Search ranking or frequency) affects the influence of a paper or repository?

- It affects papers only
- It affects repositories only
- It affects both
- It affects neither

**Q8.** What other factors do you believe may influence the visibility or influence of a benchmark work?

- Answer: ⎯⎯⎯⎯⎯⎯⎯⎯⎯⎯⎯⎯⎯⎯

SECTION B — CODE QUALITY OF BENCHMARK REPOSITORIES

**Q9.** What is the minimum level of quality you expect from a benchmark's code repository?

- High quality (almost runnable out-of-the-box)
- Basic quality (runnable after some debugging)
- No quality requirement as long as it is open-sourced
- No need to open-source
- Other: ⎯⎯⎯⎯⎯⎯⎯⎯⎯⎯⎯⎯⎯

**Q10.** What type of quality checking do you think benchmark repositories should undergo?

- Static analysis
- Manual review
- Both static analysis and manual review
- No quality checks needed
- Other: ⎯⎯⎯⎯⎯⎯⎯⎯⎯⎯⎯⎯⎯

**Q11.** When evaluating whether a repository is usable (assuming you are not required to use it), how much time do you typically spend?

- Less than 2 hours
- 2–4 hours
- 4–6 hours
- More than 6 hours

**Q12.** Would you like benchmark repositories to include a minimal runnable example (a minimal test script)?

- Yes
- Neutral
- No

**Q13.** Which of the following do you think a repository should *at least* include? (multiple choice)

- Installation guide
- Data guide
- Ethical considerations
- Other: ⎯⎯⎯⎯⎯⎯⎯⎯⎯⎯⎯⎯⎯

**Q14.** Ideally, which items should the installation or usage guide include? (multiple choice)

- Installation guide
- Data guide
- Ethical considerations
- Other: ⎯⎯⎯⎯⎯⎯⎯⎯⎯⎯⎯⎯⎯

SECTION C — COMPARISON BETWEEN BENCHMARK AND NON-BENCHMARK REPOSITORIES

**Q15.** On average, whose code quality do you believe is higher?

- Benchmark repositories
- Non-benchmark repositories
- Not sure: _________________________

**Q16.** On average, whose influence do you believe is higher?

- Benchmark papers/repositories
- Non-benchmark papers/repositories
- Not sure: _________________________

SECTION D — RELATIONSHIP BETWEEN CODE QUALITY AND INFLUENCE

**Q17.** When you decide whether to cite a paper (assuming no external requirement to reproduce the results), how does code quality affect your decision?

- Code quality does not affect whether I cite the paper
- Higher code quality increases my willingness to cite
- Higher code quality decreases my willingness to cite
- Other: _________________________

# N    SURVEY RESULTS

We conducted a 17-question anonymous survey and distributed it via public academic channels ( covering over 50 LLM safety researchers outside our institution). The survey did not mention this paper or any related context. We received 17 valid responses. Below, we summarize the primary findings.

## N.1    PAPER AND REPOSITORY INFLUENCE (Q1–Q8)

***Paper influence.*** 10/17 respondents considered *citation density* to be the most appropriate metric. 2/17 suggested "other metrics," but none provided concrete alternatives.

***Repository influence.*** For code repositories, 10/17 preferred *GitHub star density*. 4/17 noted that for *LLM models* (not applicable in this study), HuggingFace statistics can sometimes be more informative.

***Author, institution, geolocation.*** For Q3–Q5, most respondents agreed that author prominence, institutional affiliation, and geolocation affect the visibility or influence of papers and repositories.

***Publication status.*** Responses to Q6 were divided: 9/17 believed publication venue matters, whereas 8/17 believed it does not.

***Search visibility.*** For Q7, 12/17 stated that public search visibility affects *papers only*, while 5/17 believed it affects *both* papers and repositories.

***Additional factors.*** In Q8, two respondents highlighted that blog posts from major labs such as OpenAI or Anthropic can substantially influence visibility.

## N.2    CODE REPOSITORY QUALITY (Q9–Q14)

***Minimum acceptable standard (Q9).*** 14/17 selected "basic quality" (runnable after some debugging). One respondent commented that they would prefer *no release* over a poorly maintained open-source repository.

***Quality checking (Q10).*** 12/17 preferred *manual review*, and 5/17 preferred *manual review + static analysis*. No respondent favored static analysis alone.

***Time acceptable for usability checking (Q11).*** 11/17 were willing to spend $< 2$ hours; 5/17 accepted 2–4 hours; no respondent accepted more than 6 hours.

***Minimal runnable example (Q12).*** 15/17 wanted a minimal runnable example; 2/17 were neutral.

***Minimum required repository contents (Q13).*** 12/17 selected the installation guide only. 5/17 selected both installation and data guides. This indicates that installation instructions are considered the baseline component.

***Ideal repository contents (Q14).*** All 17/17 selected both installation and data guides as ideal requirements. 2/17 additionally recommended including ethical or responsible-use considerations.

### N.3 BENCHMARK VS. NON-BENCHMARK COMPARISON (Q15–Q16)

***Code quality (Q15).*** 12/17 believed benchmark repositories have higher code quality.

***Influence (Q16).*** Responses were mixed: 8/17 selected benchmark work, 4/17 selected non-benchmark work, and 5/17 were unsure. 3 of the "unsure" respondents noted that non-benchmark work has a "higher upper bound but also a very low lower bound."

### N.4 RELATIONSHIP BETWEEN CODE QUALITY AND INFLUENCE (Q17)

11/17 reported that higher code quality increases their willingness to cite a paper. 6/17 stated that code quality does not affect their citation decision. No respondent reported that higher code quality decreases their willingness to cite.

## O EXECUTION TIME

In the main text, the execution time metric for runnable benchmarks is defined as the wall-clock time from the beginning of the debugging process to the first successful completion of the official example script. Concretely, after cloning the repository, we start timing once we begin to resolve environment dependencies, install required packages, and adjust configuration files or paths as needed. The timer stops when the example script finishes successfully without errors. We denote this quantity by

$$T_{\text{clone} \to \text{finish}} = \text{Time from start of debugging to successful completion of the example script.}$$

This metric is intended to approximate the real user experience: it captures not only the intrinsic runtime of the script, but also the practical effort required to make the benchmark actually work on a fresh machine (e.g., resolving missing dependencies, fixing path issues, or updating outdated instructions). Providing a concise, lightweight quick-start example that can be executed quickly is generally considered good practice for benchmark usability, and $T_{\text{clone} \to \text{finish}}$ is designed to reflect this.

In addition, we introduce an alternative metric that explicitly separates environment setup and debugging costs from the intrinsic execution time of the example. Let $T_{\text{clone} \to \text{finish}}$ denote the wall-clock time from successful cloning of the repository to the successful completion of the example script, and let $T_{\text{exec}}$ denote the runtime of the example script itself under a correctly configured environment. We define

$$T_{\text{setup}} = T_{\text{clone} \to \text{finish}} - T_{\text{exec}}.$$

By construction, $T_{\text{setup}}$ isolates the overhead introduced by environment setup and debugging while removing confounding factors due to the script's own execution time.

For repositories labeled as *not runnable*, we find that currently all failure arises from environment setup or debugging issues (e.g., missing dependencies, incompatible package versions, broken paths), rather than from excessive demo execution time. Consequently, whether we adopt the original metric or the alternative definition, the runnability outcome for these repositories remains unchanged. For the average execution time, we report the results of the two definitions in Figure 8.

## P SUPPLEMENTARY TABLES AND FIGURES

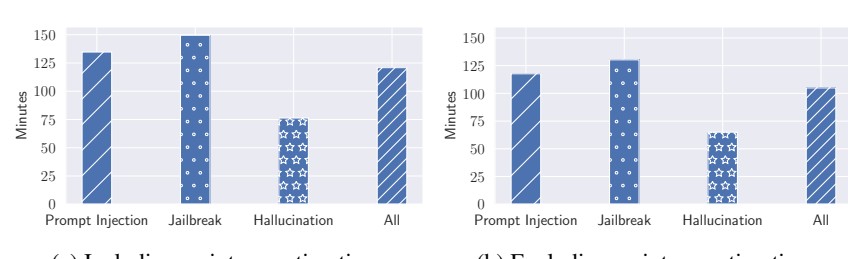

(a) Including script execution time.    (b) Excluding script execution time.

Figure 8: Average time to successfully run the example scripts in the repositories.

Table 7: Summary of the metrics used in influence evaluation.

| Dimension | Metric |
|---|---|
| Academic Community | Citation Count
Citation Density |
| Open-Source Community | GitHub Star Count
GitHub Star Density |
| Cross-Disciplinary | Scientific Field Count |

Table 8: An outline of the metrics measured during the human-based evaluation.

| Aspect | Description |
|---|---|
| Code Quality | Is the necessary dataset available in the repository?
Are the provided example scripts runnable?
Are there any extra modifications required (such as fixing bugs) besides those mentioned in the guides when trying to run the example scripts? |
| Supplementary Material Quality | Does the repository provide good install guides?
Does the repository provide useful data guides?
Does the repository contain essential ethical considerations? |

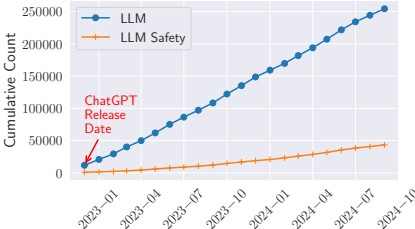

Figure 9: Cumulative papers count (from Semantic Scholar).

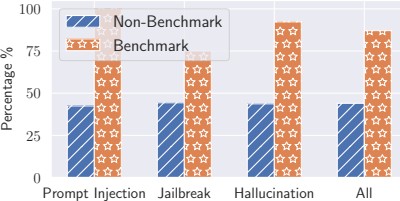

Figure 10: GitHub repository availability proportions.

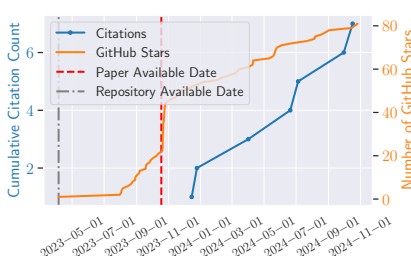

Figure 11: A typical example of the general pattern we identify. After the paper is publicly available, its GitHub star count increases rapidly.

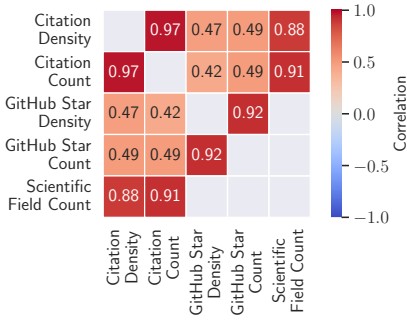

Figure 12: Spearman correlation $\rho$ matrix of the influence metrics (those with $p \geq 0.05$ are omitted).

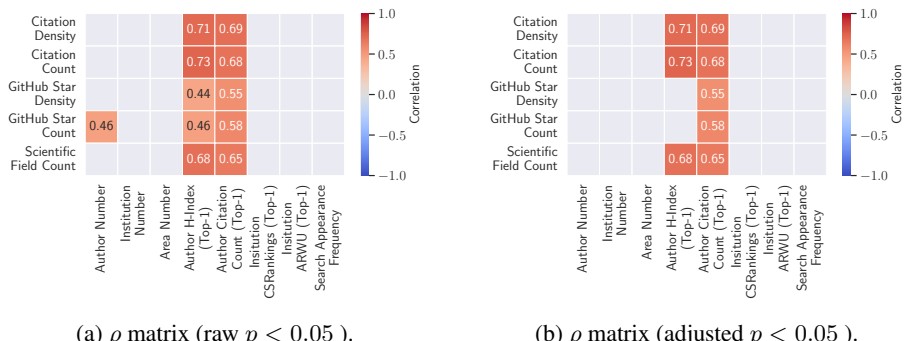

(a) $\rho$ matrix (raw $p < 0.05$ ).  (b) $\rho$ matrix (adjusted $p < 0.05$ ).

Figure 13: Spearman correlation matrices between the influence metrics and the potential quantitative factors. The unadjusted p-values on the left can be interpreted exploratively.

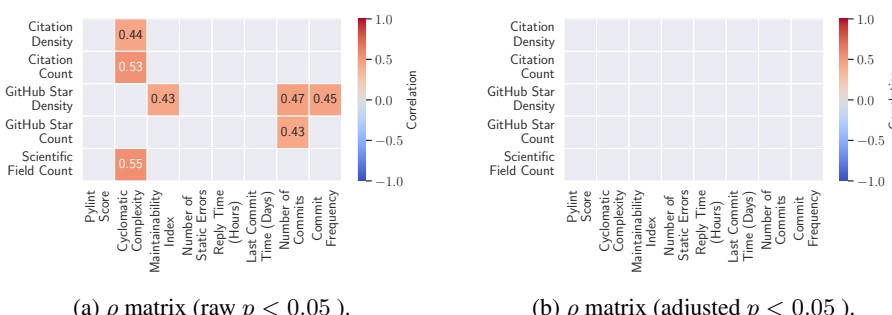

(a) $\rho$ matrix (raw $p < 0.05$ ).  (b) $\rho$ matrix (adjusted $p < 0.05$ ).

Figure 14: Spearman correlation matrices between the influence metrics and the code repository quality metrics. The unadjusted p-values on the left can be interpreted exploratively.

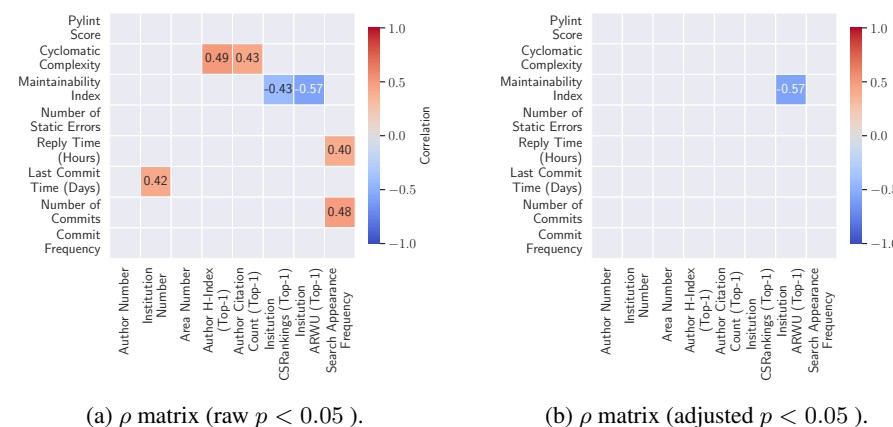

(a) $\rho$ matrix (raw $p < 0.05$ ).  (b) $\rho$ matrix (adjusted $p < 0.05$ ).

Figure 15: Spearman correlation matrices between the code repository quality metrics and the potential quantitative factors. The unadjusted p-values on the left can be interpreted exploratively.

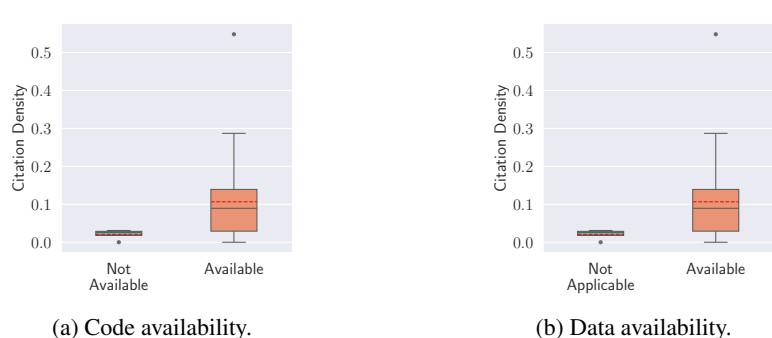

(a) Code availability.  (b) Data availability.

Figure 16: Box plots of citation density by group. The red dashed lines represent the means.

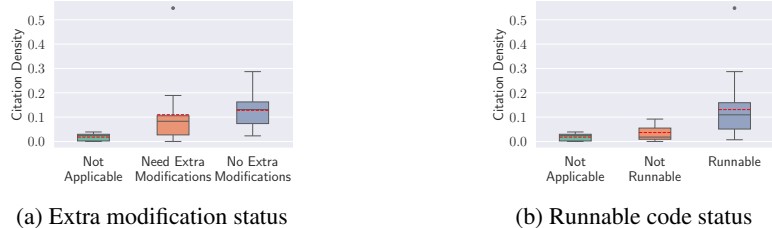

(a) Extra modification status  (b) Runnable code status

Figure 17: Box plots of citation density and the status of extra modifications and runnable code. The red dashed lines represent the mean values.

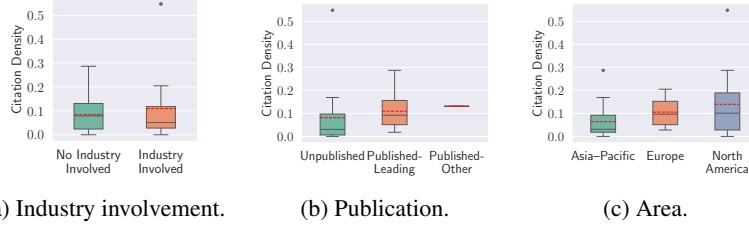

(a) Industry involvement.  (b) Publication.  (c) Area.

Figure 18: Box plots of citation density and various potential qualitative factors. The red dashed lines in the box plots represent the mean values. The further to the right the x-axis tick is, the higher the mean value of the corresponding box.

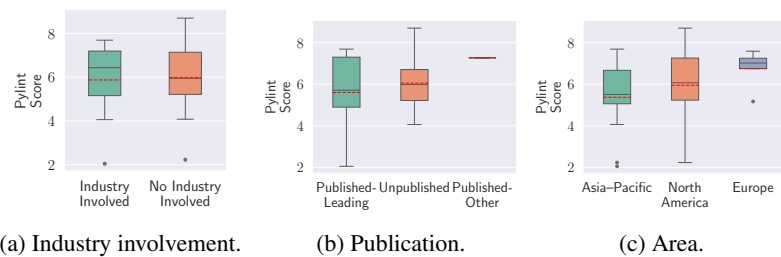

(a) Industry involvement.  (b) Publication.  (c) Area.

Figure 19: Box plots of Pylint score and various potential qualitative factors. The red dashed lines in the box plots represent the mean values. The further to the right the x-axis tick is, the higher the mean value of the corresponding box.

