# OpenReview forum: "Benchmark of Benchmarks: Unpacking Influence and Code Repository Quality in LLM Safety Benchmarks"
_ICLR.cc/2026/Conference — Submitted to ICLR 2026_

### Official Review · Reviewer_6K5m · 2025-10-27

**Soundness:** 2
**Presentation:** 3
**Contribution:** 3
**Rating:** 4
**Confidence:** 4

**Summary:**

The paper „Benchmark of Benchmarks: Unpacking Influence and Code Repository Quality in LLM Safety Benchmarks” presents the results of an analysis of 31 benchmarks on safety issues (jailbreaks, hallucinations, prompt injection). The analysis is uses 382 papers on safety issues that are not benchmarks as control group to measure difference between benchmark papers and other papers. Moreover, the authors study how several factors influence the success of benchmark papers measured in citations. The study finds that author prominence is related to success of the benchmarks, but that code quality is not. Moreover, having executable code is important, while having perfect documentation has no strong influence.

**Strengths:**

The paper „Benchmark of Benchmarks: Unpacking Influence and Code Repository Quality in LLM Safety Benchmarks” presents the results of an analysis of 31 benchmarks on safety issues (jailbreaks, hallucinations, prompt injection). The analysis is uses 382 papers on safety issues that are not benchmarks as control group to measure difference between benchmark papers and other papers. Moreover, the authors study how several factors influence the success of benchmark papers measured in citations. The study finds that author prominence is related to success of the benchmarks, but that code quality is not. Moreover, having executable code is important, while having perfect documentation has no strong influence.

**Weaknesses:**

While most of the paper seems sound, there is a general weakness in the setup of the statistics (comment 1) and, crucially, an important confounding factor that is not considered at all that could also explain one core conclusion, i.e., the relationship between benchmark success and author prominence (comment 1). For me, this second point is the most crucial aspect and the main reason for my judgment. The comments from 3 onwards are relatively minor.
1) I wonder why the analysis is not based on a linear model. The coefficients of a linear model effectively measure linear correlations, same as Pearson’s correlation coefficient. However, a linear model has the additional advantage that correlation between factors can be taken into account, typically leading to better results when multiple factors are analyzed in parallel.
2) While I intuitively also believe that author prominence is related to success, I do not believe that the study design is sufficient to conclude this. Notably, nothing in the study design controls *benchmark quality* (e.g., size of benchmark, novelty of benchmark, quality of benchmark data, or similar). Since this is not controlled for, there is a simple, possible alternative explanation for the observed effect that is not ruled out: prominent authors create author higher-quality benchmarks. The lack of control for this confounder is the key issue I have with this study that greatly limits the value of this conclusion.
3) The execution time until the example scripts were run is a strange and biased metric. If a benchmark is more comprehensive, this might have longer examples, which would not be a bad thing. Still, this would show up as higher execution time. A cleaner measurement would be to measure the time between cloning the repository and starting the successful run, since this would only quantify the researcher effort to get to this point, excluding the confounding effect of example runtime.
4) The ethics section is not an ethics section at all. Instead, the section reports limitations of the study. This needs to be changed and format requirements must be adhered to.
5) Figure 2 is very hard to read and I strong suggest to avoid this strange pie-bar like visualization method. A grouped (stacked?) bar chart would likely be a lot easier to read. This is actually what is used in Figure 3, which is easier to read. However, Figure 3 suffers from bad legend placement, which partially hides the information.
6) There seems to be a mismatch between table 2 and the textual description of the contents (Page 4, L181). The text mentions the geolocation, which I cannot find in the table. The table instead mentions area, which I assume is the research area – though I may misinterpret this. Anyways, this should be harmonized.
7) All references are broken (missing brackets), likely because the latex stile was changed without ever checking this.

**Questions:**

1) What is the impact of not using a linear model for the analysis, especially given that there seem to be strong correlations between the independent variables?
2) Why are the results valid, even though there is no control for actual benchmark quality?

---

> ### Author Response · Authors · 2025-11-24
> **Reply 1**
>
> We sincerely thank the reviewer for the careful reading and the very insightful and constructive feedback. In the following, we address each raised point in detail and hope our responses and the accompanying revisions fully clarify the concerns.
>
> >W-1, Q-1. I wonder why the analysis is not based on a linear model. The coefficients of a linear model effectively measure linear correlations, same as Pearson’s correlation coefficient. However, a linear model has the additional advantage that correlation between factors can be taken into account, typically leading to better results when multiple factors are analyzed in parallel.
> What is the impact of not using a linear model for the analysis, especially given that there seem to be strong correlations between the independent variables?
>
> **(1)** Why not linear modeling?
>
> We fully agree that the multiple linear regression (MLR) model is theoretically good at controlling for confounding variables.
> However, it requires a larger sample size.
> The total population of relevant benchmark papers is inherently small (N=31),  while our multiple linear model contains 8 predictor variables.
> According to statistical guidelines, approximately 10 observations per predictor are needed to obtain stable estimates in regression [1][2][3].
> With such a small sample size relative to the number of predictor variables, the regression coefficients become highly unstable, the variance inflates, and the risk of overfitting becomes significant.
> Furthermore, there might be considerable correlation among our predictors (multicollinearity), which further reduces the reliability of the estimated coefficients in the small N setting.
> Therefore, multiple linear regression may be statistically underpowered here.
>
> Paired Pearson correlation analysis, on the other hand, has relatively lower sample size requirements, and our sample size meets the requirements, resulting in a more robust outcome in this case.
>
> We still very much appreciate the reviewer’s insight. To address this point, we have added two exploratory regression models (with citation count as the dependent variable, all variables have been standardized) to Appendix L.
>
> **(2)** What is the impact of not using a linear model for the analysis?
>
> We believe the impact is limited in terms of our reported findings.
>
> Based on our two additional linear regression models, we find that the MLR results are generally consistent with our main findings from the Pearson correlation. Specifically, in the MLR analysis, the code quality metric remains statistically insignificant (p > 0.05). The author's top-1 h-index also shows a positive correlation with the citation count (p = 0.002).
>
> At the same time, the MLR coefficients seem to show instability which may be caused by multicollinearity under small N, such as large coefficient magnitudes (closed to 1) and coefficient sign reversals between related predictors (i.e., the model assigned a significant negative coefficient to the "maximum author citation count" (-0.80, p = 0.01) while assigning a very large positive coefficient to the "h-index" (0.98, p = 0.002)).
> These behaviors indicate that MLR is not that reliable to serve as a primary analytic tool in our setting, even though its overall directional conclusions align with the more conservative Pearson analysis.
>
> Taken together, we believe the current approach based on Pearson analysis strikes the balance between statistical rigor and data limitations, and the supplementary MLR results corroborate—rather than contradict—our main conclusions.
> Therefore, the impact of not using MLR as the primary method is minor, and our conclusions remain robust. We hope these clarifications address the reviewer’s concern.
>
> [1] Statsols. A Guide to Sample Size for Regression Models. StatSolS.
> [2] Pripp A H et al. Sample size for a prediction model. Tidsskrift for Den norske legeforening. 2024.
> [3] Portland State University. Sample Size and Power for Regression.

---

> ### Author Response · Authors · 2025-11-24
> **Reply 2**
>
> >W-2, Q-2.
> While I intuitively also believe that author prominence is related to success, I do not believe that the study design is sufficient to conclude this. Notably, nothing in the study design controls benchmark quality (e.g., size of benchmark, novelty of benchmark, quality of benchmark data, or similar). Since this is not controlled for, there is a simple, possible alternative explanation for the observed effect that is not ruled out: prominent authors create author higher-quality benchmarks. The lack of control for this confounder is the key issue I have with this study that greatly limits the value of this conclusion.
> Why are the results valid, even though there is no control for actual benchmark quality?
>
> We sincerely thank the reviewer for raising this thoughtful and important concern.
> We fully acknowledge the conceptual possibility that author prominence (A) might affect benchmark quality (C), which could in turn affect influence metrics (B).
> Below, we clarify why our conclusions remain valid and statistically grounded, and we will revise the manuscript accordingly to avoid any misunderstanding.
>
> **(1)** Our analysis focuses on observable statistical associations rather than causal inference.
>
> Our findings (based on statistical evidence) state that author prominence (A) is statistically associated with citation density (B). We do not claim that A causally determines B. Therefore, even if a causal chain A → C → B exists, this would still imply an association between A and B, which means our findings still exist.
>
> To avoid misinterpretation, we will revise the manuscript to more explicitly frame our results as correlational rather than causal.
>
> **(2)** All papers in our dataset satisfy a minimum threshold of scientific quality.
>
> Every benchmark paper we analyzed originates from arXiv or comparable preprint platforms, each of which applies basic screening on scientific content, formatting, ethical norms, and completeness. In addition, all papers appear in widely used, actively maintained community repositories.
> Our manual checking process did not reveal any obviously low-quality manuscripts.
> This means that extremely low-quality papers are naturally excluded, and the lower tail of “paper quality” is already truncated.
> The papers' quality could be considered to be in a limited, practical range.
> While we acknowledge that distinguishing exceptionally high-quality papers is beyond our current methodological capabilities, we will add this limitation to the manuscript (in the new limitation section).
>
> **(3)** We greatly appreciate that the reviewer’s comment touches on a fundamental challenge in meta-research: the lack of a widely accepted metric for intrinsic paper quality.
>
> We agree with the reviewer that controlling for C (intrinsic scientific quality) would be ideal. In fact, the reviewer has identified a core pain point that the bibliometrics and meta-science communities have struggled with for decades.
>
> To our knowledge, no generally accepted, objective, scalable measure of “paper quality” exists. Prior meta-studies typically rely on coarse proxies (e.g., page length, abstract length) [4][5][6] or rare experimental designs (e.g., submitting the same article to different venues) [7], which have well-known limitations. If a reliable and domain-agnostic quality metric could be developed, it would significantly improve both meta-research and peer-review efficiency.
>
> Within these methodological constraints, we have made our best effort to operationalize observable aspects of benchmark “quality” by using measurable proxies such as automated code-quality metrics and human-run reproducibility tests. Importantly,
> our statistical analyses find no significant association between these proxies and author prominence. If the reviewer can suggest any concrete, quantifiable, and scalable measure of intrinsic benchmark quality suitable for this context, we would be very happy to incorporate it into our analysis.
>
> In summary, we thank the reviewer again for the advice.
> We will clarify the correlational nature of our findings, explicitly acknowledge the limitations in measuring intrinsic paper quality, and revise the manuscript so that the scope and interpretation of our results are fully transparent to readers.
>
> [4] Xie J et al. The correlation between paper length and citations: a meta-analysis. Scientometrics. 2019.
> [5] Falagas M E et al. The Impact of Article Length on the Number of Future Citations: A Bibliometric Analysis of General Medicine Journals. PLoS One. 2013.
> [6] Mammola S et al. Measuring the influence of non-scientific features on citations. Scientometrics. 2022.
> [7] Larivière V et al. The impact factor’s Matthew Effect: A natural experiment in bibliometrics. Journal of the American Society for Information Science and Technology. 2010.

---

> ### Author Response · Authors · 2025-11-24
> **Reply 3**
>
> >W-3.Time to measure
>
> We thank the reviewer for the valuable feedback.
> We agree that “the time for the example script to finish running” may be affected by the inherent runtime of the benchmark script itself.
> We provide several clarifications below, which we hope will address your concerns.
>
> **(1)** From the perspective of benchmark design and usability, a user-friendly codebase typically provides a concise, lightweight quick-start example rather than a large-scale script that runs on the full dataset. Providing an overly large or time-consuming example is itself considered not user-friendly. In our survey, we also found that 15/17 expect minimal runnable examples.
>
>  Therefore, in the main text, we adopt $T_{\text{clone} \rightarrow \text{finish}}$ (the total time from debugging to successful completion of the example script) as an evaluation metric, with the intention of reflecting the actual user experience that benchmark authors generally aim to provide.
>
> **(2)** In our manual inspection of all benchmarks, the vast majority of repositories indeed provide small demos that run quickly using only a small number of data samples. For repositories marked as “not runnable,” the execution logs show that all failures were caused by environmental dependencies or configuration errors. There were no cases where the script failed to run due to long execution time. Thus, script runtime does not affect runnability in our study, and our main conclusions remain unaffected.
>
> **(3)** Regarding the metric you proposed—“the time between cloning the repository and starting the successful run”—we fully understand the motivation behind it.
>
> To avoid ambiguity in its definition, we operationalize it as:
> $T_{\text{setup}} = T_{\text{clone} \rightarrow \text{finish}} - T_{\text{exec}}.$
> *(The time from successful cloning to successful completion of the example script, minus the execution time of the script itself.)*
>
> This revised metric isolates environment setup and debugging costs while removing confounding factors introduced by the execution time of the benchmarking example.
>
> After excluding script execution time, we found that the average time $T_{\text{setup}}$ on all runnable benchmarks decreased to about 105 minutes ($T_{\text{clone} \rightarrow \text{finish}}$ is about 120 minutes). This is still a fairly long time, almost reaching the tolerance limit of most survey respondents (11/17). Our main findings are also not affected much.
> We have added this metric to Appendix O, and we will explicitly mention this in the revision.
>
> >W-4.Ethics statement
>
> We appreciate the reviewers’ valuable comments on the ethics statement.
> We thank one reviewer for describing the original statement as “an example to follow,” particularly noting its detailed discussion of methodological limitations. At the same time, we fully understand the concern raised by another reviewer, namely that combining ethical considerations with methodological limitations may reduce the focus of a traditional ethics statement.
> The ICLR Code of Ethics [8][9] covers a broad range of responsible research practices, including transparency about research limitations, which may naturally lead to different interpretations of what should be included in the ethics statement. We respect both perspectives.
>
> To resolve this and improve clarity for readers, in the revised version, we:
> - retain the positively noted ethical components
> - move methodological limitations to a dedicated “Limitations” section
> - refocus the ethics statement strictly on ethical risks, dual-use concerns, privacy considerations, and responsible research practices, aligning it with the conventional structure of ethics statements.
>
> >W-5. Readability of Figures 2 & 3
>
> We have updated the PDF accordingly. The legends are now placed uniformly above each figure to avoid obstruction.
>
> >W-6. Area
>
> “Area” refers to the geographic region. We have updated the terminology consistently in the revised PDF. Additional details are provided in Appendix H.
>
> >W-7.Citation format
>
> We have corrected the citation formatting accordingly.
>
> [8] https://iclr.cc/public/CodeOfEthics.
> [9] https://iclr.cc/Conferences/2026/AuthorGuide.
>
> We are deeply grateful for the reviewers’ careful reading and constructive suggestions. We hope that our detailed responses and the substantially revised manuscript adequately address the raised concerns. Please don’t hesitate to let us know if there is anything else we can clarify or improve — we truly appreciate your time and effort.

---

> > ### Comment · Reviewer_6K5m · 2025-11-25
> >
> > Thank you for the detailed response. I appreciate the experiment with the linear model and understand the concern with the sample size. However - to ask bluntly - are you not rather pointing out that your data size is too small to draw statistically sound conclusions about this many factors, because you lack a sufficient amount of data?
> >
> > Moreover, while I appreciate that the correlation measurement is valid even if the causal relationship is different, this is not what (most) readers will understand, nor what is stressed with the finding of the Matthews effect. So while this is acknowledged in the limitations, the paper still strongly implies that prominence is the driving factor. Thus, my key concern remains.
> > Here are a couple of examples for why I believe this is problematic. ImageNet comes from a well-known influential research group. But the benchmark itself was groundbreaking and is, thus highly cited. Another example for LLMs is SWE-Bench. One of the GPT authors is involved. However, the benchmark is not highly cited because of that, but rather because it provided a very good and new benchmarking framework.
> >
> > And I could list more.
> >
> > Regarding a suggestion for criteria, I also cannot easily provide them. I would, however, submit that “it is on arXiv” is far away from a reasonable criterion that establishes a similar level of scientific quality and importance. Publication venues could at least somewhat mitigate this (e.g., an oral at an A* conf gets more exposure than a random preprint) and this was at least judged as high-quality and high-impact by peer-reviewers. However, looking at subgroups per venue type (and acceptance type?!) probably again reduces the amount of data too much, leading to similar problems as with the linear model.
> >
> > Thus, my concern regarding the soundness remains unresolved. I understand all underlying issues and understand that they are somewhere between hard and impossible to solve.  If the threat were negligible, adding this as limitation would be sufficient for me. But I do not think it is, as this control has a big impact on the conclusion, which could be two ways:
> > - Either prominent authors have more influential papers/benchmarks *because* the papers are good.
> > - Or prominent authors have more in influential papers/benchmarks *because* they are influential.
> >
> > The truth is probably be somewhere in the middle, but my point is: this paper does not help us to understand which it is (to which degree). The correlation analysis and wording strongly implies the second aspect. And that is my (unresolved) core concern.

---

> > > ### Author Response · Authors · 2025-11-26
> > > **Thanks for your quick reply (Reply 1).**
> > >
> > > Thanks for your quick reply and further explanation. Please see our clarification below.
> > >
> > > **(1) Sample Size**
> > >
> > > Thank you for the reviewer’s detailed comments.
> > > We fully understand your concerns regarding the sample size and statistical robustness.
> > > Our position is: the current sample size is insufficient to support a robust multivariate linear regression (MLR), but for the Pearson pairwise correlation analysis we use, its statistical significance is adequate.
> > > The two types of analyses have inherently different sample size requirements.
> > > The specific explanations are as follows:
> > >
> > > 1. Statistical power and overfitting:
> > > For 8 potential factors and 31 observations, the sample-to-predictor ratio of a multivariate linear regression model (MLR) is about 3.8:1. This is far below the commonly recommended minimum ratio of 10:1 or 15:1. Forcing an 8-factor regression on this dataset would result in insufficient degrees of freedom, leading to overfitting and unstable coefficients.
> > >
> > > 2. Effectiveness of the current method:
> > > However, for the specific purpose of identifying correlations, Pearson correlation analysis remains statistically valid and robust for this sample size (usually N>=25 provides sufficient statistical power to reliably detect pairwise associations [1][2], and we have N=31).
> > >
> > > 3. Data constraints in this field:
> > > Our dataset ($N = 31$) represents a comprehensive survey of current LLM safety benchmarks, rather than a small-scale random sample. At present, no larger sample group exists.
> > >
> > > In summary, the current limited sample size does not affect the statistical validity of the main observational conclusions we obtained based on pairwise Pearson correlations.
> > > We have added Appendix L.1 to further explain this choice.
> > >
> > > [1] Francis McIntyre et al. Tables of the Ordinates and Probability Integral of the Distribution of the Correlation Coefficient in Small Samples. Mathematics. 2008.
> > > [2] Minitab Support. Data considerations for Correlation. Minitab. 2024.

---

> ### Author Response · Authors · 2025-11-26
> **Thanks for your quick reply (Reply 2).**
>
> **(2) Clarification of Correlations and Causal Relationships**
>
> We fully understand your concerns, as well as the potential misunderstandings that may arise among most readers.
> And we sincerely appreciate the reviewer’s understanding that, under the current limitations in this emerging area, this causal disentanglement is indeed difficult—and in some ways impossible—to resolve at this stage.
>
> Therefore, we add a new footnote to emphasize that our findings concern only correlation and cannot be used to infer causality.
> In addition, we completely removed the content related to the “Matthew Effect in Science” from the main text.
> We list the potential reasons for these correlations only in the footnote, including the Matthew effect, the intrinsic quality of benchmarks, and so on.
> We now ensure that we report only objective statistical data and findings derived from statistical measures, without including potential subjective speculations.
>
> Detailedly, we do the following modifications:
>
> We revise Section 4.3 (line 258-262) and add a footnote to avoid misleading:
> >We also observe statistical associations between several author-prominence metrics and influence metrics. Specifically, moderate positive correlations exist between Author H-Index (Top-1) and several influence metrics, with r of 0.38 (citation count), 0.41 (GitHub star count), and 0.45 (scientific field count). Additionally, Author Citation Count (Top-1) shows stronger correlations with GitHub Star Count (0.65) and GitHub Star Density (0.62).
>
> Footnote in Section 4.3:
> >Note: We only report correlations, not causality. These correlations do not imply that author prominence must drive influence. The correlations may stem from multiple factors, such as the “Matthew Effect in Science” (Merton, 1968) (where reputation amplifies attention), intrinsic higher work quality of prominent scholars, or their interplay. Disentangling such confounding factors is currently beyond our scope and capability.
>
> We revise the Section 9 Conclusion (line 533-535):
> >We observe statistical associations between certain author-prominence measures and influence metrics.
> However, we do not detect statistically significant correlations between those author-prominence measures and code-quality indicators.
>
> We revise the main findings in Section 1:
> >• (Section 4) Benchmark papers do not show a statistically significant difference in citation metrics compared to non-benchmark papers, but their code repository quality tends to be higher. Statistical associations are observed between several author-prominence metrics and influence metrics, for example, between Author H-Index (Top-1) and citation count.
> • (Section 5) Both the code quality and the supplementary material quality have considerable room for improvement: only 39% of code repositories can run smoothly without modifications, only 16% provide flawless install guides, and only 6% include ethical considerations. Author prominence has no statistically significant correlation to code quality.
> • (Section 6) Simply providing code is not enough: we find no statistically significant difference in citation density between papers with code requiring modifications and those without any code. We also fail to find a significant correlation between a paper’s citation density and the intrinsic code characteristics, such as static metrics or maintenance frequency.
>
> We believe that the current revision under your advice will not lead to reader misunderstanding, and enable them to more clearly understand the findings of this work, including our measurements of benchmark papers’ influence and code quality, as well as the deficiencies and areas for improvement in the code quality of existing benchmark papers.
>
> Thank you for your suggestions. We hope these modifications have addressed your concerns, and we welcome any further feedback or questions.

---

> > ### Comment · Reviewer_6K5m · 2025-11-27
> >
> > Thanks for the additional clarifications and update. One final question, regarding the sample sizes:
> >
> > While I agree that for a *single* feature, your argument is valid, for multiple features you have a similar issue as with multiple regression as results might become random due to repeated testing. A possible simple way to resolve (or confirm) my concerns is to compute the p-values of the (individual) correlation coefficients and check for their significance, accounting for repeat tests (e.g., Bonferroni-Holm)?
> >
> > With this data, you could show that your results are likely not explained by a random effect as a consequence of the (from a statistical point of view) small sample size.

---

> > > ### Author Response · Authors · 2025-11-28
> > > **Thanks for your quick reply.**
> > >
> > > Thank you for your prompt response and valuable advice.
> > > Based on our understanding, we think your last concern has two aspects: data size and repeated tests.
> > > The Bonferroni-Holm adjustment you kindly suggested targets repeated tests.
> > > We will discuss each of these aspects in turn.
> > >
> > > **(1) Dataset Size**
> > >
> > > We replaced the original Pearson analysis with Spearman correlation analysis, which is insensitive to outliers and has lower data requirements.
> > > Spearman is particularly suitable for our scenario with a small full sample size. [1][2]
> > >
> > > Given the limited dataset size, standard significance testing based on asymptotic theory poses a risk of invalidity.
> > > To mitigate this, we conduct permutation tests to calculate empirical $p$-values.
> > > This approach generates a simulated distribution by shuffling the data 10,000 times, allowing us to determine significance directly from the observed data structure rather than theoretical approximations.[3][4][5]
> > >
> > > Through these two steps, we believe that we have adequately addressed the limitations of small full-sample cases under the existing circumstances.
> > >
> > > **(2) Repeated Tests**
> > >
> > > To account for repeated tests,  following your kind advice, we applied the Bonferroni-Holm adjustment to control for multiple comparisons within each dependent variable.
> > > This step strictly controls the Family-Wise Error Rate (FWER).
> > > To maintain rigorous standards, we only report results with corrected $p$-values $< 0.05$ as statistically significant in the main text.
> > > We also report the $\rho$ of raw (unadjusted) $p$-values in the Appendix.
> > > These unadjusted results should be interpreted only as exploratory signals requiring further validation in larger cohorts.
> > >
> > > **The main findings are still valid.**
> > > After the two steps described above, we found that the main conclusions remained valid and unchanged.
> > > For example:
> > > - Strong positive monotonic correlations exist between {Author H-Index (Top-1)} and several influence metrics, with $\rho$ of 0.73 (Citation Count) and 0.71 (Citation Density).
> > > The p-values ​​before adjustment were 1.9998e-04 and 3.9996e-04, respectively.
> > > The p-values ​​after adjustment were 3.1997e-03 and 6.3994e-03, respectively.
> > > - The author’s h-index (top-1) and citation count (top-1) show no significant correlation with code quality indicators($ p_\text{adjusted} > 0.05 $)
> > >
> > > **Revision of the PDF**.
> > > In addition, we have updated all relevant content in the revision PDF, with the most important updates including:
> > > - Section 4.3 line 253-259
> > > >We observe statistical monotonic correlations between several author-prominence metrics and influence metrics.
> > > Specifically, strong positive correlations exist between {Author H-Index (Top-1)} and several influence metrics, with $\rho$ of 0.73 (Citation Count), 0.71 (Citation Density), and 0.68 (scientific field count).
> > > Additionally,{Author Citation Count (Top-1)} shows stronger correlations with{GitHub Star Count} (0.58) and {GitHub Star Density} (0.55).
> > >
> > > - Section 5.5 line 411-417
> > > >While high-influence authors are often associated with popular benchmarks, their involvement \textbf{is not necessarily associated with} higher code quality.
> > > {The author’s h-index (top-1) and citation count (top-1) show no significant correlation with code quality indicators($ p_\text{adjusted} > 0.05 $)}.
> > > However, we observe a strong negative correlation between ARWU ranking and code maintainability index ($ \rho = -0.57 $).
> > > These suggest that higher-ranked institutions are associated with more maintainable code.
> > >
> > > - Section 6 line 436-440
> > > >None of the influence metrics are observed to have a significant correlation with code quality indicators ($ p_\text{adjusted} > 0.05 $), suggesting that code following a higher coding standard is not necessarily linked to more citations.
> > >
> > > - Appendix L.1.
> > > Now we additionally discuss the choice of Spearman, the permutation test, and the Bonferroni-Holm correction.
> > >
> > > [1] Phipson & Smyth, "Permutation P‑values Should Never Be Zero: Calculating Exact P‑values When Permutations Are Randomly Drawn," Stat. Appl. Genet. Mol. Biol. (2010).
> > > [2] Holm, "A Simple Sequentially Rejective Multiple Test Procedure," Scand. J. Stat. (1979).
> > > [3] Good, Permutation, Parametric, and Bootstrap Tests of Hypotheses, Springer (2005).
> > > [4] Edgington & Onghena, Randomization Tests, CRC Press (2007).
> > > [5] Ernst, "Permutation methods: A basis for exact inference," Stat. Sci. (2004).
> > >
> > > We hope we have fully addressed your concerns and still welcome your further questions.

---

### Official Review · Reviewer_g9kb · 2025-10-30

**Soundness:** 1
**Presentation:** 2
**Contribution:** 2
**Rating:** 2
**Confidence:** 4

**Summary:**

This paper evaluates the academic influence and code quality of 31 LLM safety benchmarks compared to 382 non-benchmark papers. The authors claim that benchmark papers show no clear advantage in citations, and neither author prominence nor paper influence correlates with code quality. Many repositories have usability and ethical shortcomings, with only 39% of repositories ready-to-use and 6% addressing ethical considerations. The authors suggest that prominent researchers should take the lead in improving standards.

**Strengths:**

The paper presents several interesting and valuable findings.

Notably, some of the ethical and reproducibility-related metrics—such as only 39% of repositories being ready-to-use, 16% including flawless installation guides, and a mere 6% addressing ethical considerations—highlight the need for researchers to pay more attention to open-sourcing and maintaining their code alongside their research contributions.

Additionally, the observation that author's h-index does not show a strong correlation with code quality is intriguing and provides an important perspective on the relationship between academic influence and research artifacts.

**Weaknesses:**

Much of the experimental design seems to rely on somewhat imprecise metrics and a fair amount of manual inspection, which makes the paper’s motivation a bit hard to follow.

Additionally, the conclusions, experimental design, and motivation appear somewhat subjective, reflecting the authors’ own perspective rather than broader community evidence. It might be more appropriate to frame this part as an initial motivation supported by larger-scale community surveys rather than just the authors’ judgment. For example, the statement in the introduction, “counterintuitively, we find that benchmark papers show no significant advantage in academic influence over non-benchmark papers”, carries some subjective interpretation; collecting feedback from a larger set of participants could make this claim more robust and less reliant on individual judgment.

**Questions:**

1.	Relevance of conclusions in RQ1: The authors conclude that benchmark papers do not show a statistically significant difference in citation metrics compared to non-benchmark papers, based on GitHub Citation Count and Citation Density. However, benchmark and non-benchmark papers inherently serve different roles in research, so it is not clear that a “higher or lower” comparison is meaningful. Additionally, measuring influence through GitHub stars may be misleading: benchmarks often serve as tools that are widely used but not necessarily “starred,” whereas other papers may receive stars more readily. Therefore, using GitHub data alone to assess influence may introduce bias.

2.	Experimental design and evaluation metrics: Although the authors acknowledge limitations in the “Imperfect Metrics” section of the ethics statement, the choice of metrics seems questionable. For instance, in RQ2, the primary criterion for evaluating benchmark quality is code quality and reproducibility. This seems unusual, because benchmarks are meant to provide measurement standards in a field, and the underlying evaluation ideas may be more important than the engineering quality of the code. Many researchers are not professional software engineers, and their code is often written for research purposes rather than for production-grade usage. While such criteria may make sense for toolboxes, it is unclear why code quality should be the main standard for benchmarks. This seems closer to evaluating whether research code undergoes proper code review rather than assessing the scientific quality of the benchmark itself.

3.	Subjectivity of conclusions: While the paper’s conclusions may have value, the path to reaching them appears questionable. Determining how to evaluate research work, what metrics best reflect quality, and the relative importance of code within a research contribution are highly subjective issues. More justification or broader evidence may be needed to support the chosen evaluation design.

---

> ### Author Response · Authors · 2025-11-24
> **Reply 1**
>
> Thank you for your effort in reviewing the manuscript.
> Below is our response to your questions.
> We hope they can resolve your misunderstandings and concerns.
>
> >Q1. Relevance of conclusions in RQ1: The authors conclude that benchmark papers do not show a statistically significant difference in citation metrics compared to non-benchmark papers, based on GitHub Citation Count and Citation Density. However, benchmark and non-benchmark papers inherently serve different roles in research, so it is not clear that a “higher or lower” comparison is meaningful. Additionally, measuring influence through GitHub stars may be misleading: benchmarks often serve as tools that are widely used but not necessarily “starred,” whereas other papers may receive stars more readily. Therefore, using GitHub data alone to assess influence may introduce bias.
>
> (1) While we appreciate the reviewer’s concern that benchmark papers and non-benchmark papers may play different conceptual roles, comparing the scholarly impact of different categories of contributions is a standard and well-established practice in infometrics and meta-research. Prior work routinely conducts cross-category comparisons (e.g., reviews vs. original research [1], domain-relevant vs. methodologically relevant papers [2]) to examine whether different forms of contributions exhibit systematic differences in influence.
>
> Our RQ1 does not assume that one category “should” receive more citations than the other; it poses a purely empirical question: within the same field, do benchmark papers and non-benchmark papers differ statistically in citation-based influence metrics? Our analysis indicates that they do not. As such, the comparison is methodologically legitimate and consistent with the established infometrics tradition. Furthermore, based on the results of our survey Q16, the question is also currently unclear.
>
> (2) We fully agree that no single metric, including “GitHub stars,” is an ideal measure of influence. This is a widely acknowledged limitation and open challenge in scientometrics: almost all practical studies rely on imperfect yet informative proxy indicators [3][4][5][6][7]. For this reason, our study intentionally incorporates five metrics, including Citation Count, Citation Density, Scientific Field Count, GitHub Star Count, and GitHub Star Density, to mitigate the weaknesses of any single indicator.
>
> Regarding the reviewer’s hypothesis that benchmarks may be widely used but less likely to be starred, this is an interesting possibility; however, it does not appear in our data—benchmark repositories do not show systematically lower star densities, instead, they show higher.
>
> Finally, our community survey similarly indicates that researchers do not currently agree on a clearly superior alternative metric for repository influence. This suggests that GitHub-based metrics, while imperfect, remain a practical and commonly accepted proxy.
>
> [1] Miranda R et al. Overcitation and overrepresentation of review papers in the most cited papers. J Informetr. 2018;12(4):1015-1030.
> [2] Verma M K et al. Scientometric assessment of funded scientometrics and bibliometrics research (2011–2021). Scientometrics. 2023;128:4305-4320.
> [3] York University Libraries. Limitations of bibliometrics. York University Libraries: Research Metrics & Visibility; [n.d.].
> [4] Borgman C L. Big Data, Little Data, No Data: Scholarship in the Networked World. Cambridge/MA: MIT Press; 2015.
> [5] Moed H F. Citation Analysis in Research Evaluation. Dordrecht: Springer; 2005.
> [6] Bornmann L et al. Scientometrics in a changing research landscape: Bibliometrics has become an integral part of research quality evaluation and has been changing the practice of research. EMBO Rep. 2014;15(12):1228-1232.
> [7] Thelwall M. Web Indicators for Research Evaluation: A Practical Guide. Springer; 2016.

---

> ### Author Response · Authors · 2025-11-24
> **Reply 2**
>
> >Q2. Experimental design and evaluation metrics: Although the authors acknowledge limitations in the “Imperfect Metrics” section of the ethics statement, the choice of metrics seems questionable. For instance, in RQ2, the primary criterion for evaluating benchmark quality is code quality and reproducibility. This seems unusual, because benchmarks are meant to provide measurement standards in a field, and the underlying evaluation ideas may be more important than the engineering quality of the code. Many researchers are not professional software engineers, and their code is often written for research purposes rather than for production-grade usage. While such criteria may make sense for toolboxes, it is unclear why code quality should be the main standard for benchmarks. This seems closer to evaluating whether research code undergoes proper code review rather than assessing the scientific quality of the benchmark itself.
>
> We sincerely thank the reviewer for the insightful observation in Q1 that “benchmarks function more like tools.” We fully agree with this perspective, and it directly motivates the design logic behind RQ2.
>
> First, we would like to clarify that this study does not treat “code quality” as the core scientific quality criterion of a benchmark, nor does it attempt to evaluate benchmark papers in terms of task design, metric construction, or scientific contribution.
> The scope of RQ2 is independent and clearly circumscribed: it focuses solely on the quality of benchmark code repositories, namely, whether the benchmark, as a tool on which the research community relies, can be executed and reused smoothly.
>
> Within this scope, we define “code repository quality” as a composite framework consisting of two complementary components:
>
> 1. Usability dimensions that directly affect practical use.
> These include whether example scripts can be executed, whether additional modifications are required, whether installation guides and data guides are complete, and whether essential ethical considerations are provided. These dimensions are assessed through systematic human evaluation and reflect the practical challenges researchers may encounter when using a benchmark in real-world research scenarios (see Figure 2 and Figure 3 in the paper).
>
> 2. Structural indicators derived from static analysis.
> These include static errors, code complexity, and maintainability. Although these indicators originate from software engineering tools, they are highly relevant to benchmark usability in practice: static errors often correspond to non-executable code paths, and excessive complexity typically indicates structures that are difficult to modify or reuse. Thus, in this study, static analysis is not only to measure engineering style, but also to serve as an objective automated signal that helps identify potential usability obstacles.
>
> Based on this design, “code repository quality” in our study should be understood as a multi-dimensional combination of “usability + static structural characteristics.” This framework does not concern a benchmark’s scientific merit, theoretical design, or task validity. Rather, it evaluates only the practical usability of the benchmark as a tool.
>
> In summary, our research question is **not** “Is this benchmark scientifically superior?”, but an independent and more focused one: "What is the quality of the benchmark’s code repository?"
> We believe that this circumscribed and code-quality–focused perspective is fully consistent with the reviewer’s characterization of benchmarks as tools and helps reveal the practical bottlenecks faced by current LLM safety benchmarks.

---

> ### Author Response · Authors · 2025-11-24
> **Reply 3**
>
> >Q3. Subjectivity of conclusions: While the paper’s conclusions may have value, the path to reaching them appears questionable. Determining how to evaluate research work, what metrics best reflect quality, and the relative importance of code within a research contribution are highly subjective issues. More justification or broader evidence may be needed to support the chosen evaluation design.
>
> Thanks for your comment.
> Below, we clarify why our evaluation design does not rely on subjective judgment and is fully aligned with established scientometrics and meta-research practices.
>
> **(1)** Our metrics are clearly defined, widely used, and supported by prior literature [1][2][3], and we intentionally adopt a diverse multi-metric framework.
>
> All indicators employed in our study, including citation density, GitHub star density, scientific field count, static analysis metrics (Pylint score, cyclomatic complexity, maintainability index), maintenance-frequency metrics, and runnable-code status, are objectively measurable quantities with unambiguous operational definitions.
> These metrics have been extensively used across scientometrics, software engineering, and reproducibility studies.
> Furthermore, to avoid over-reliance on any single imperfect proxy, we intentionally employ a diverse set of complementary metrics, following the standard practice in scientometrics.
> As discussed in our Ethics Statement, imperfect metrics are intrinsic to the field; the accepted norm is to use transparent, reproducible proxies rather than subjective assessments.
>
> **(2)** All conclusions are derived strictly from statistical evidence rather than subjective interpretation.
>
> We rely exclusively on statistical tests such as Mann–Whitney U for distributional comparison and Cliff’s delta for effect size quantification. Every comparative conclusion in the manuscript follows directly from these statistical outputs. No finding is based on the authors’ beliefs, impressions, or manual interpretation. Moreover, we make no causal claims; all statements strictly report whether the observed data show statistically significant differences or correlations.
>
> **(3)** We appreciate the reviewer’s suggestion and will revise the manuscript accordingly.
>
> To avoid any potential misinterpretation, we will remove expressions such as “counterintuitively” and ensure all phrasing remains strictly descriptive and aligned with the statistical evidence.
> We thank the reviewer for helping us improve clarity and neutrality.
>
> **(4)** We also thank the reviewer for suggesting broader community evidence; our external survey supports our metric choices.
>
> As supplementary evidence, we conducted a 17-question anonymous survey among LLM-safety researchers outside our institution (17 valid responses).
> A majority of respondents considered metrics such as “citation density” and “GitHub star density” appropriate for measuring influence. We emphasize that this survey serves only as supplementary confirmation.
> Our metric choices are grounded in established scientometrics practice, and we appreciate the reviewer’s comment, which encouraged us to make this connection more explicit.
> The full content related to the survey is available in Appendix M and Appendix N.
>
> [1] Jones R et al. Citation analysis of the 100 most common articles regarding distal radius fractures. J Clin Orthop Trauma. 2017.
> [2] Sandison A. Patterns of citation densities by date of publication in Physical Review. J Am Soc Inf Sci. 1975;26(6):349-352.
> [3] Kadic A J et al. Research methodology used in the 50 most cited articles in the field of pediatrics: types of studies that become citation classics. BMC Med Res Methodol. 2020.
>
> We genuinely hope that the revised manuscript and our responses satisfactorily address the reviewers’ previous concerns.
> Should any further clarification be needed, we would be more than happy to provide additional details and explanation.

---

### Official Review · Reviewer_jdUQ · 2025-10-31

**Soundness:** 4
**Presentation:** 4
**Contribution:** 4
**Rating:** 6
**Confidence:** 5

**Summary:**

This paper presents an study to evaluate the quality of benchmarks of large language models (LLMs) and safety.  The authors investigated three research questions (RQs): RQ1:the influence of current benchmark RQ2: the quality associated code repositories of the benchmark and factors to assess the quality; and RQ3: the relationship between influence of benchmark papers and the code quality.

**Strengths:**

Strengths
- Timely topic focusing on the safety of LLM benchmark
- Broader impact towards the research community and well as industry practioners who deploy or develop LLM

**Weaknesses:**

Weakness.
- Any insights of security researchers would be insightful

**Questions:**

N/A

---

> ### Author Response · Authors · 2025-11-24
> **Thanks for your review.**
>
> We appreciate a lot your positive feedback on our paper and recognition of meta-studies. Please see our response below regarding the weakness.
>
> >Weaknesses: Any insights of security researchers would be insightful
>
> Thanks for your insightful advice.
> To incorporate insights from external security researchers, we designed a 17-question anonymous survey and distributed it via public academic channels ( covering over 50 LLM safety researchers outside our institution) without mentioning this paper or any related context.
> We have received 17 valid responses so far.
> The major findings are summarized below and are included in the revised version.
>
> 1. Paper and repository Influence.
> Q1: 10/17 respondents believe citation density is a better measure of academic influence. 2/17 suggested “other metrics” but were unable to propose clearly better alternatives.
> Q2: For code repositories, 10/17 chose GitHub star density. 4/17 noted that for LLM models (not applicable in our study), HuggingFace metrics can sometimes be more relevant.
> Q3–Q5: Most respondents stated that author, institution, and geolocation all correlate with the influence of papers/repositories.
> Q6: Responses were split—9/17 believe publication status matters, 8/17 believe it does not.
> Q7: 12/17 believe public search visibility is related to papers only; 5/17 believe it is related to both papers and repositories.
> Q8: Two respondents explicitly mentioned OpenAI/Anthropic blog posts as an additional channel that is associated with the influence.
>
> 2. Code repository quality.
> Q9 (minimum acceptable standard): 14/17 selected “basic quality” (runnable after some debugging). One respondent wrote that they would prefer no release over low-quality open-sourcing.
> Q10 (quality checking): 12/17 preferred manual review; 5/17 preferred manual + static analysis.
> Q11 (acceptable time to test if the code repository is runnable): 11/17 tolerate <2 hours, 5/17 tolerate 2–4 hours; no one accepts >6 hours.
> Q12 (minimal runnable example): 15/17 want such an example; 2/17 are neutral.
> Q13 (minimum required contents of a repository): When asked about the minimum components a repository must include:
> 12/17 only selected the installation guide. 5/17 selected both the installation guide and data guide. (This indicates installation instructions are viewed as the baseline requirement.)
> Q14 (ideal contents of a repository): When asked what contents repositories ideally should include, all 17/17 selected both the installation guide and the data guide; 2/17 additionally selected ethical considerations. (This shows broad consensus on what high-quality repositories should provide.)
>
> 3. Benchmark vs. non-benchmark comparison.
> Q15: 12/17 believe benchmark repositories have higher code quality.
> Q16: Influence evaluations were mixed—8/17 chose benchmark, 4/17 chose non-benchmark, and 5 were unsure.
> 3 of those choosing “unsure” noted that non-benchmark papers have “a higher upper bound but also a very low lower bound.”
>
> 4. Relationship Between Code Quality and Citations
> Q17: 11/17 stated that higher code quality increases their willingness to cite the corresponding paper; 6/17 reported that code quality does not affect their citation decision.
>
> The full survey content and results are reported in Appendix M and Appendix N of the revised PDF.

---

### Official Review · Reviewer_xYHc · 2025-10-31

**Soundness:** 4
**Presentation:** 4
**Contribution:** 3
**Rating:** 8
**Confidence:** 4

**Summary:**

The paper presents an analysis of LLM safety benchmarks, analyzing 31 benchmarks and 382 non-benchmarks. The analyses what causes benchmark papers to get cited. Is it code quality or author prominence? It also analyses wheter the benchmark papers are cited more than non-benchmark papers.

Data is colleced in a structured and transparent way.  The analysis is well-motivated and rigorous relying on statistical analyses.

Conclusions are prominent resarchers are cited more, code quality is not important, benchmark papers are not more cited than non-benchmark papers (at least in this subfield), and becnhmark with functional code that can be used without modification are more cited than those offering code that requires modifications

**Strengths:**

* The paper is well written and structured well.
* The methodology is rigorous, and I trust the conclusions.
* The conclusions are interesting and, I would guess, probably valid in general and not only for LLM safety benchmarks, although none that have ever looked at resarch code will be surprised to learn that resarch code often is not of high quality (the incentives are not there). Everyone that publishes a benchmark should take note that making it easy to run increasesits scientific impact.
* The ethical statement is very nice and an example to follow with its detailed discussion of limitations of the method.

**Weaknesses:**

I do not find many weaknesses in this study. On the contrary, I find it very rigorous and trustworthy.

My main concern is whether the topic is too narrow, as it is a benchmark of LLM safety benchmarks. It is a bit on the side of representation learning, so I am not sure the community would value this study despite its many strong qualities.

My strong belief is that this type of meta studies that informs us, the AI community, what constitutes good resarch and what influences impact are important. However, I am well aware that many do not.

**Questions:**

Did you consider to follow the PRISMA methodology [1] for structured literature reviews when conducting your structured search for benchmarks?

I would have liked to see a flow diagram, such as the one proposed in the PRISMA methodology, for understanding how many papers were retrieved and excluded at different steps.

[1] https://www.prisma-statement.org/

---

> ### Author Response · Authors · 2025-11-24
> **Thanks for your review.**
>
> We are very grateful for your recognition of the meta-study field and our work. Below are our responses to your questions and weaknesses.
>
> >Questions
>
> Thank you for the helpful suggestion.
> Following your recommendation, we have added a PRISMA-style flow diagram that details the identification, screening, and exclusion steps of our benchmark collection process.
> The new description is now included in Appendix F, and the flow diagram is presented in Figure 4.
> We believe this addition improves the transparency of our methodology.
>
> >Weaknesses
>
> Thank you very much for your kind recognition of the rigor of our work, and especially for your willingness to advocate for greater attention in the academic community to meta-studies of this kind.
> We fully understand and share your concern that research focused on LLM safety benchmarks may sometimes be viewed as too specialised and not valued enough by the community.
>
> We believe it is important to stress that LLM safety is not a marginal topic: recent incidents, such as the use of ChatGPT to assist in planning a vehicle explosion in Las Vegas [1], demonstrate how AI systems, and the benchmarks used to evaluate them, can have significant real-world safety implications.
> In this context, benchmarks for LLM safety play a crucial role in shaping how the community assesses, trusts, and deploys these systems.
>
> Meta-studies remain relatively rare at venues like ICLR, despite their importance for reflecting on and improving scientific practice. We sincerely appreciate the reviewer’s call for the community to value such introspective and foundational work.
> We hope that our study contributes to strengthening the empirical basis of LLM safety research and aligns with the reviewer’s vision for broader recognition of the value of meta-research in our field.
>
> [1] "Man who exploded Cybertruck in Las Vegas used ChatGPT in planning, police say", The Associated Press, https://www.npr.org/2025/01/07/nx-s1-5251611/cybertruck-explosion-las-vegas-chatgpt-ai

---

### Author Response · Authors · 2025-11-24
**Key Changes in Response to Reviewer Comments**

We are deeply grateful to the reviewers for their thorough and insightful feedback, which has significantly helped us improve the manuscript. In response to the valuable comments received, we have made the following substantial revisions:

1. New Section 8: Discussion and Limitations
We have added a dedicated Section 8 entitled “Discussion and Limitations”.
In the Discussion part, we now include the discussion about open challenges in benchmarks and a detailed discussion of the newly added survey.
In the Limitations part, we have substantially expanded the discussion on the methodological limitations (which were kindly acknowledged by the reviewers in the previous round), and further elaborated on concerns regarding scientific quality and the imperfection of current evaluation metrics.

2. Five new appendices.
We have added five new appendices to provide greater transparency and reproducibility:
Appendix F: PRISMA Flow Diagram.
Appendix L: Exploratory Multiple Linear Regression Analysis.
Appendix M: LLM Safety Benchmark Survey.
Appendix N: Survey Results.
Appendix O: Execution Time.

3. Revised Ethics Statement.
The Ethics Statement has been rewritten to fully conform to the narrow definition of an ethical considerations statement.

4. Careful rephrasing throughout the paper.
Following the reviewers’ helpful comments on interpretation, we have revised the entire manuscript to consistently adopt a purely correlational perspective.
All potentially misleading terms suggesting causality (e.g., “enhance”) have been removed or rephrased to prevent any misinterpretation as causal claims.

5. Improved Figure 2 and Figure 3.
Both figures have been redesigned for significantly better readability and visual clarity, as kindly suggested.

6. Minor fixes.
We have corrected reference formatting, typos, and other minor issues throughout the manuscript.

We sincerely hope these revisions, made in direct response to the reviewers’ thoughtful and constructive suggestions, have strengthened the paper.

---

> ### Author Response · Authors · 2025-11-28
> **Key Changes in Response to Reviewer Comments -- Continued 1**
>
> We have addressed the concerns regarding dataset size and repeated testing.
>
> First, to account for the small sample size, we replaced the original Pearson correlation with Spearman correlation, which is robust to outliers and suitable for limited data, and further conducted permutation tests (10,000 iterations) to obtain empirical p-values.
>
> Second, following the suggestion, we applied the Bonferroni-Holm adjustment to control for repeated tests, reporting results with significant corrected p-values in the main text, while including results with significant raw p-values in the Appendix for exploratory purposes.
>
> Our main findings remain unchanged.
> All relevant sections and the Appendices have been updated.

---

### Author Response · Authors · 2025-12-03
**Summary of Discussion**

We thank the AC and reviewers for their participation in the ICLR review process.
Below, we provide a summary of the entire review and rebuttal.

---

We conducted the first multi-dimensional evaluation of LLM safety benchmarks, assessing their impact and code quality.
We found that benchmark papers do not exhibit significant advantages over non-benchmark papers in terms of academic impact.
We identified a critical misalignment: while author prominence correlates with paper impact, neither author prominence nor paper impact shows significant correlation with code quality.
Our findings also indicate substantial room for improvement in code and supplementary materials.

---

Our work was recognized by reviewers as ***well-structured*** (Reviewer xYHc), ***with interesting and valuable findings*** (Reviewers g9kb, 6K5m), and having ***broader impact towards the research community and well as industry practioners who deploy or develop LLM*** (Reviewer jdUQ).
In particular, Reviewer xYHc noted that ***this type of meta studies that informs us, the AI community, what constitutes good research and what influences impact are important.***

---

We have addressed the reviewers' questions in detail:

**Reviewer xYHc** (original rating: 8):
- Following their suggestion, we added a PRISMA-style flow diagram to enhance the readability and clarity of the manuscript.

**Reviewer jdUQ** (original rating: 6):
- We added an external questionnaire survey to gather broader community perspectives.

**Reviewer 6K5m** (original rating: 4):
We engaged in multiple rounds of discussion with the reviewer.
  - Through revisions to the manuscript wording, legends, figures, clarifications of certain definitions, and additional exploratory linear model experiments, Reviewer 6K5m acknowledged that we addressed most of their concerns.
  - When we responded to the reviewer's final concern regarding dataset size and repeated testing, the ICLR discussion period was frozen.
Using Spearman correlation analysis (for dataset size), permutation tests for p-values (for dataset size), and Bonferroni-Holm adjustment (for repeated testing), we confirmed that our main conclusions remain unchanged.
We believe we have solved the reviewer's final concern.

These analyses have been incorporated into the manuscript.
We think that we have fully addressed all of Reviewer 6K5m's concerns, including the final one.

**Reviewer g9kb** (original rating: 2):
- We clarified Reviewer g9kb's factual misunderstandings of our work:
(1) The reviewer's subjective assumptions were refuted by our statistical data.
(2) We never claimed what the primary evaluation criteria for benchmarks should be.
(3) All our conclusions are based on objective statistical data or analysis.
(4) Our methodology follows established conventions for meta-studies.
- In addition, we collect more feedback from the community by distributing the survey.

We believe our clarifications resolve Reviewer g9kb's misunderstandings of our work.

---

We have incorporated all revisions into the latest version of the manuscript.
We believe that through our additional clarifications, explanations, and experiments, we have addressed all of the reviewers' concerns.

Finally, we once again thank the AC for their extra effort during the review process and the reviewers for their participation in the ICLR review. We believe our work contributes to a deeper understanding and improvement of LLM safety benchmarks.

Best regards,

Authors

---

### Meta-Review · Area_Chair_ASfC · 2026-01-08

**Summary:**

This is an unusual paper about community meta analysis and reflection. It analyzed works in the safety domain, dividing them into benchmarks vs non-benchmarks and raised some general questions about the quality of work in this area. For example:  while author prominence correlates with paper influence, but no significant correlation with code quality. They show room for improvement in code and supplementary materials: only 39% of repositories are ready-to-use, 16% include flawless installation guides, and a mere 6% address ethical considerations.

Review scores (2468) are very mixed. With a rating of 8, reviewer xYHc raised the most important point that:
"My main concern is whether the topic is too narrow, as it is a benchmark of LLM safety benchmarks. It is a bit on the side of representation learning, so I am not sure the community would value this study despite its many strong qualities."

The two positive reviewers mostly show they appreciate this style of work, rather than enthusiastic endorsement of this particular work. Reviewer jdUQ is rather uninformative. The two negative reviewers raised mostly technical problems with the methodology that are not fully addressed.

Given the reviews and the nature of this paper, this paper clearly should not be accepted. While the community should welcome the occasional self-reflection with the support of the reviewers, such work should be of strong general relevance to the community. This work is too specific to safety benchmarks and the main recommendation that prominent researchers should do better work and increase their code quality is not an interesting reflection even if all the technical methods and claims well-supported.

**Reviewer Concerns:**

main concerns are not addressed, some technical concerns are addressed

**Reviewer Scores:**

probably not

---

### Decision · Program_Chairs · 2026-01-26

Reject